# TAB: Temporal Accumulated Batch Normalization in Spiking Neural Networks

**Haiyan Jiang**[1]    **Vincent Zoonekynd**[2]    **Giulia De Masi**[3,4]    **Bin Gu**[1,6*]    **Huan Xiong**[1,5*]

[1]Mohamed bin Zayed University of Artificial Intelligence, UAE

[2]Abu Dhabi Investment Authority, UAE   [3]Technology Innovation Institute, UAE

[4]Sant'Anna School of Advanced Studies, Italy   [5]Harbin Institute of Technology, China

[6]School of Artificial Intelligence, Jilin University, China

`haiyan.jiang@mbzuai.ac.ae`  `vincent.zoonekynd@adia.ae`
`giuliademasi@gmail.com`  `{bin.gu, huan.xiong}@mbzuai.ac.ae`

## Abstract

Spiking Neural Networks (SNNs) are attracting growing interest for their energy-efficient computing when implemented on neuromorphic hardware. However, directly training SNNs, even adopting batch normalization (BN), is highly challenging due to their non-differentiable activation function and the temporally delayed accumulation of outputs over time. For SNN training, this temporal accumulation gives rise to Temporal Covariate Shifts (TCS) along the temporal dimension, a phenomenon that would become increasingly pronounced with layer-wise computations across multiple layers and multiple time-steps. In this paper, we introduce TAB (Temporal Accumulated Batch Normalization), a novel SNN batch normalization method that addresses the temporal covariate shift issue by aligning with neuron dynamics (specifically the accumulated membrane potential) and utilizing temporal accumulated statistics for data normalization. Within its framework, TAB effectively encapsulates the historical temporal dependencies that underlie the membrane potential accumulation process, thereby establishing a natural connection between neuron dynamics and TAB batch normalization. Experimental results on CIFAR-10, CIFAR-100, and DVS-CIFAR10 show that our TAB method outperforms other state-of-the-art methods.

## 1 Introduction

Spiking Neural Networks (SNNs) are known to be biologically inspired artificial neural networks (ANNs) and have recently attracted great research interest (Chowdhury et al., 2022; Ding et al., 2022). The attraction of SNNs lies in their ability to deliver energy-efficient and fast-inference computations when implemented on neuromorphic hardware such as Loihi (Davies et al., 2018) and TrueNorth (Akopyan et al., 2015; DeBole et al., 2019). These advantages arise from the fact that SNNs utilize spikes to transmit information between layers, whereby the networks circumvent multiplication during inference (Roy et al., 2019). However, the discrete and non-differentiable nature of the binary firing functions makes it difficult to directly train deep SNNs. ANN-to-SNN conversion (Diehl et al., 2015; Bu et al., 2022; Jiang et al., 2023) and directly training with surrogate gradients back-propagation (Neftci et al., 2019; Deng et al., 2022; 2023) are two typical solutions.

Batch Normalization (BN) has found extensive use in ANNs and has seen tremendous success in boosting their performance by reducing the internal covariate shift (ICS) and flattening the loss landscape (Ioffe & Szegedy, 2015; Santurkar et al., 2018). In ANNs, ICS refers to changes in the distribution of layer inputs caused by updates of preceding layers, while in SNNs, the *Temporal Covariate Shift (TCS) phenomenon* (Duan et al., 2022) has been identified due to updates of preceding layers and prior time-steps, which transpires along the additional temporal dimension. Within SNNs, synaptic currents are sequentially fed into spiking neurons, with spike-triggered asynchronous currents accumulating in the membrane potential. Whenever this accumulated membrane potential exceeds a threshold, a spike is generated. This *temporal dependency* on membrane accumulation

---

*Corresponding authors. Codes are available at `https://github.com/HaiyanJiang/SNN-TAB`.

has the potential to amplify the internal covariate shift across the temporal domain. The intertwining of this *temporal dependency with the TCS phenomenon*, presents a significant challenge in direct training of SNNs especially for the integration of BN techniques into SNNs.

When it comes to BN techniques for SNNs, only a few methods have been proposed. These methods either normalize data jointly by aggregating data across the temporal dimension or perform independent normalization at each discrete time-step. For example, Kim & Panda (2021) conducts independent batch normalization separately at each time-step. However, this approach uses separate sets of mean, variance, and scale and shift parameters at each time-step, failing to account for the temporal dependencies of the input spikes. While Zheng et al. (2021) merges the data along the time dimension and utilizes shared batch statistics across all time-steps for normalization. Nonetheless, introducing such overall statistics may limit the flexibility to capture varying temporal characteristics at different time-steps. On the other hand, Duan et al. (2022) attempts to tackle the TCS issue by assigning different weights to each time-step, while still utilizing shared batch statistics across all time-steps for normalization. Although these methods improve upon the performance of the SNN models, they do not significantly address the alignment with the neuron dynamics, i.e., the membrane accumulation dependency, or provide a potential to do so.

In this paper, we propose TAB (Temporal Accumulated Batch Normalization) as a solution to effectively address these challenges by closely aligning with the neuron dynamics, specifically the accumulated membrane potential, and providing more accurate batch statistics. This alignment establishes a natural connection between neuronal dynamics and batch normalization in SNNs. Neuron dynamics refer to the changes in the membrane potential of a neuron over time as it integrates input signals and generates spikes. Here, "aligning with neuron dynamics" means that TAB is tailored to mimic or capture neurons' behavior as closely as possible, normalizing data in line with the temporal dependencies and information accumulation within neurons. This alignment ensures that TAB's normalization process corresponds well with how neurons naturally operate in SNNs, thus leading to improved performance by addressing the temporal covariate shift problem.

## 2 BACKGROUND

### 2.1 RELATED WORK

**SNN Learning Methods.** Many works have recently emerged and focused on the supervised training of SNNs (Wu et al., 2021a; Zhou et al., 2021; Meng et al., 2022; Xiao et al., 2021). These SNN learning methods can be mainly categorized into two classes: ANN-to-SNN conversion (Diehl et al., 2015; Deng & Gu, 2021; Ding et al., 2021; Han et al., 2020; Li et al., 2021a; Bu et al., 2022; Hao et al., 2023; Lv et al., 2023) and end-to-end training with back-propagation (Fang et al., 2021a; Zhang & Li, 2020; Deng et al., 2022; Xiao et al., 2022; Guo et al., 2022; Meng et al., 2023). ANN-to-SNN conversion takes a pre-trained ANN and converts it into an SNN by preserving the weights and replacing the ReLU activation function with a spiking activation function. This approach can be efficient in obtaining an SNN since the ANN has already been trained and the weights can be directly copied to the SNN. However, the resulting performance of the converted SNN may not be as good as that of the original source ANN. It usually requires a large number of time-steps for the converted SNN to achieve performance comparable to the source ANN. Direct end-to-end training usually employs the surrogate gradients (Wu et al., 2018; 2019; Neftci et al., 2019; Zheng et al., 2021; Eshraghian et al., 2021) method to overcome the non-differentiable nature of the binary spiking function to directly train SNNs from scratch. This method can yield comparable performance to that of traditional ANNs with a few time-steps.

**BN Method in ANNs.** Batch normalization methods have significantly contributed to the success of ANNs by boosting their learning and inference performance (Ioffe & Szegedy, 2015; Xiong et al., 2020; Bjorck et al., 2018). BN is a technique used to stabilize the distribution (over a mini-batch) of inputs to each network layer during training. This is achieved by introducing additional BN layers which set the first two moments (mean and variance) of the activation distribution to zero and one. Then, the batch-normalized inputs are scaled and shifted using learnable/trainable parameters to preserve model expressiveness. This normalization is performed before the non-linearity is applied. The BN layer can be formulated as,

$$\text{BN}(x_i) = \gamma \hat{x}_i + \beta \,, \quad \hat{x}_i = \frac{x_i - \mu}{\sqrt{\sigma^2 + \epsilon}} \,, \quad i = 1, \cdots, b \,.$$

The mini-batch mean $\mu$ and variance $\sigma^2$ are computed by $\mu = \frac{1}{b}\sum_{i=1}^{b} x_i$ and $\sigma^2 = \frac{1}{b}\sum_{i=1}^{b}(x_i-\mu)^2$.

**BN Method in SNNs.** Due to the additional temporal dimension, several recent studies have proposed modifications to batch normalization to fit the training of SNNs. The threshold-dependent Batch Normalization (tdBN) method (Zheng et al., 2021) is introduced to alleviate the gradient vanishing or explosion during training SNNs. The tdBN utilizes shared BN statistics and parameters (as the conventional BN) by merging the data along the temporal dimension. Similar to tdBN, the TEBN method (Duan et al., 2022) employs shared BN statistics by merging the data along the temporal dimension, then scales using different weights to capture temporal dynamics. Different from them, BNTT (Kim & Panda, 2021) uses separate BN statistics and parameters at each time-step $t$ independently, however, it ignores the temporal dependencies of the input spikes. Differently, our TAB method leverages the accumulated pre-synaptic inputs in the temporal domain, which is in alignment with the membrane potential accumulation in the LIF model.

## 2.2 SPIKING NEURON DYNAMICS AND NEURON MODEL

SNNs use binary spike trains to transmit information between layers. Each neuron maintains its membrane potential dynamics $\boldsymbol{u}_i(t)$ over time, "integrates" the received input with a leakage (much like an RC circuit), and fires a spike if the accumulated membrane potential value exceeds a threshold. We adopt the widely used leaky-integrate-and-fire (LIF) model. Neuron dynamics refer to the changes in the membrane potential of a neuron over time as it integrates input signals and generates spikes, which can be formulated as a first-order differential equation (ODE),

$$\text{LIF Neuron Dynamics:} \quad \tau\frac{d\boldsymbol{u}_i(t)}{dt} = -\boldsymbol{u}_i(t) + R\boldsymbol{I}_i(t), \quad \boldsymbol{u}_i(t) < V_{th}, \tag{1}$$

where $\boldsymbol{I}_i(t)$ is the injected input current to the $i$-th neuron at time $t$, $\boldsymbol{u}_i(t)$ is the membrane potential of the $i$-th neuron at time $t$ in the current layer, $V_{th}$ is the membrane threshold, and $\tau$ denotes the membrane time constant, and $R$ denotes the resistor. For numerical simulations of LIF neurons, we consider a discrete version of the neuron dynamics. Similar to Wu & He (2018), the membrane potential $\boldsymbol{u}_i[t]$ of the $i$-th neuron at time-step (discrete) $t$ is represented as:

$$\boldsymbol{u}_i[t] = \lambda\boldsymbol{u}_i[t-1] + \sum_{j\in\text{pre}(i)} W_{ij}\boldsymbol{o}_j[t]. \tag{2}$$

We adopt a simple current model $R\boldsymbol{I}_i[t] = \sum_{j\in\text{pre}(i)} W_{ij}\boldsymbol{o}_j[t]$, with $R$ absorbed in weights $W_{ij}$. Here, $\boldsymbol{o}_i[t]$ denotes the binary spike of neuron $i$ at time-step $[t]$, taking a value of 1 when a spike occurs and 0 otherwise. The index $j$ refers to pre-synaptic neurons. The membrane potential $\boldsymbol{u}_i[t]$ increases with the summation of input spikes from all the pre-synaptic neurons $\text{pre}(i)$ connecting the current $i$-th neuron through synaptic weight $W_{ij}$. It also decreases with a leak factor $\lambda$ ($0 < \lambda \leqslant 1$), where $\lambda$ and the time constant $\tau$ are related by $\lambda = e^{-\frac{\Delta t}{\tau}}$. The discrete LIF model degenerates to the IF model when $\lambda = 1$, therefore in the following, we only use the LIF model with $0 < \lambda \leqslant 1$. When the neuron's membrane potential $\boldsymbol{u}_i[t]$ exceeds the threshold $V_{th}$, the neuron will fire a spike with $\boldsymbol{o}_i[t] = 1$ and then reset the membrane potential to 0. By combining the sub-threshold dynamics Eq. (2) and hard reset mechanism, the whole iterative LIF model can be formulated by:

$$\text{Discrete LIF Neuron Model:} \quad \boldsymbol{u}_i[t] = \lambda\boldsymbol{u}_i[t-1](1-\boldsymbol{o}_i[t-1]) + \sum_{j\in\text{pre}(i)} W_{ij}\boldsymbol{o}_j[t], \tag{3}$$

$$\boldsymbol{o}_i[t] = H(\boldsymbol{u}_i[t] - V_{th}), \tag{4}$$

where $H(x)$ is the Heaviside step function, i.e., the non-differentiable spiking activation function. $H(x) = 1$ if $x > 0$ and $H(x) = 0$ otherwise.

## 3 PROPOSED TAB METHOD

In this section, we will present our TAB method. We begin by introducing the Temporal Dependencies and Temporal Covariate Shift in SNNs which motivate our method. Following this, we introduce our TAB method, which addresses these challenges. Finally, we establish a theoretical connection between the neural dynamics and the TAB method by deriving the closed-form solution of LIF dynamics ODE.

### 3.1 MOTIVATION: TEMPORAL DEPENDENCIES AND TEMPORAL COVARIATE SHIFT

Temporal dependencies in SNNs arise naturally from the sequential nature of spike events, where synaptic currents (also known as spike trains) are sequentially fed into spiking neurons, playing a pivotal role in capturing the dynamic evolution of input spikes over time. These networks model the dynamics of biological neurons through ODEs and utilize spikes to transmit information (Eshraghian et al., 2021). In SNNs, each neuron maintains a membrane potential, continuously 'integrating' and accumulating received spikes over time. It emits a spike only when its accumulated membrane potential exceeds a threshold, remaining inactive otherwise in the current time-step (Li et al., 2021a). This process highlights the intrinsic influence of temporal dynamics on the temporally delayed accumulation of the membrane potential. We refer to this accumulation dependency over the time dimension as temporal dependencies.

In SNNs, a phenomenon known as Temporal Covariate Shift (TCS) has been identified (Duan et al., 2022), which represents ICS (Internal Covariate Shift) (Ioffe & Szegedy, 2015) across the additional temporal dimension, and it refers to changes in the distribution of layer inputs caused by updates of preceding layers, and prior time-steps. Within the framework of SNNs, synaptic currents are sequentially fed into spiking neurons, and spike-triggered asynchronous currents are accumulated into the membrane potential which will trigger a spike when it exceeds the membrane threshold. This temporal dependency on membrane potential accumulation intensifies the internal covariate shift along the temporal domain. This temporal dependency, together with the TCS phenomenon, presents a significant challenge when integrating BN techniques into SNNs.

Our motivation comes along these lines, how to perform batch normalization in training of SNNs, but keeping in mind the temporal dependency of the data, as well as the temporal covariate shift. A simple, yet elegant, method that aligns closely with this underlying neuron dynamics comes with Temporal Accumulated Batch normalization (TAB). Generally speaking, our TAB method addresses the temporal covariate shift issue by aligning with the inherent temporal dependencies in SNNs. Fig. 1 illustrates the temporal dependencies and neuron dynamics and showcases the involvement of our proposed TAB method.

Neuronal dynamics refers to the change in membrane potential over time as a neuron integrates input signals and generates spikes. This temporal accumulation of the membrane potential in SNNs enables neurons to process input data by taking into account both past and current time-steps (with no access to future information beyond $t$), and the TAB method aligns closely with this underlying neuron dynamics and alleviates the TCS issue.

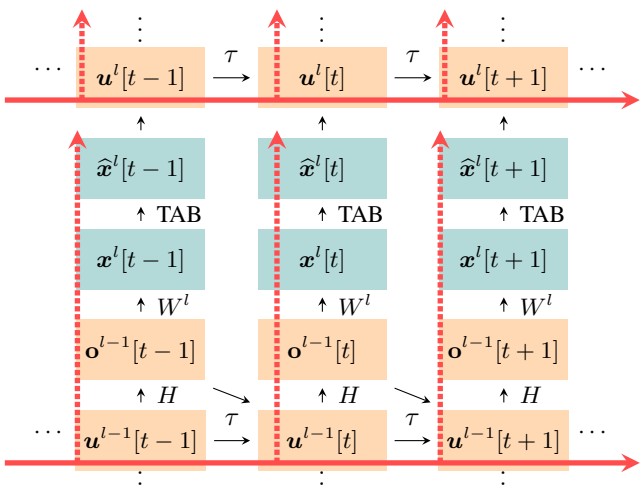

Figure 1: The temporal dependencies and neuron dynamics in SNNs, specifically the temporal dependency associated with the accumulation of membrane potential in the discrete LIF model. The black arrows represent the temporally delayed accumulation over time, while the red arrows indicate the information flow along the spatial layers (vertical axis) and the temporal domain (horizontal axis).

### 3.2 TEMPORAL ACCUMULATED BATCH NORMALIZATION (TAB)

To address the temporal covariate shift issue and to model the temporal distributions in SNNs, our TAB method aligns with the inherent temporal dependencies by utilizing the temporal accumulated batch statistics $(\mu_{1:t}, \sigma_{1:t}^2)$ over an expanding window $[1, t]$. To achieve this, we establish the relationship between the expectations and variances across accumulated time-steps $(\mu_{1:t}, \sigma_{1:t}^2)$ and those of the

single time-step $(\mu[t], \sigma^2[t])$, as follows:

$$\mu_{1:t} = \frac{1}{t} \sum_{s=1}^{t} \mu[s] \,, \quad \sigma_{1:t}^2 = \frac{1}{t} \sum_{s=1}^{t} \sigma^2[s] \,. \tag{5}$$

Our proposed TAB method utilizes Temporal Accumulated Statistics $(\mu_{1:t}, \sigma_{1:t}^2)$ for data normalization, and then assigns different learnable weights $\omega[t] > 0$ to each time-step to distinguish their effect on the final result. The TAB method is given by

$$\hat{x}_i[t] = \text{TAB}(x_i[t]) = \omega[t] \left( \gamma[t] \frac{x_i[t] - \mu_{1:t}}{\sqrt{\sigma_{1:t}^2 + \epsilon}} + \beta[t] \right) = \hat{\gamma}[t] \frac{x_i[t] - \mu_{1:t}}{\sqrt{\sigma_{1:t}^2 + \epsilon}} + \hat{\beta}[t] \,, \; \omega[t] > 0 \,. \tag{6}$$

Given the pre-synaptic inputs $\boldsymbol{x}^l[t]$ to layer $l$ at time-step $t$, the spiking neuron with TAB is as follows,

$$\boldsymbol{x}^l[t] = \boldsymbol{W}^l \boldsymbol{o}^{l-1}[t] \,, \tag{7}$$

$$\boldsymbol{u}^l[t] = \lambda \boldsymbol{u}^l[t-1](1 - \boldsymbol{o}^l[t-1]) + \widehat{\boldsymbol{x}}^l[t] \,, \tag{8}$$

$$\text{where} \quad \widehat{\boldsymbol{x}}^l[t] = \text{TAB}(\boldsymbol{x}^l[t]) = \widehat{\boldsymbol{\gamma}}[t] \frac{\boldsymbol{x}^l[t] - \boldsymbol{\mu}_{1:t}}{\sqrt{\boldsymbol{\sigma}_{1:t}^2 + \epsilon}} + \widehat{\boldsymbol{\beta}}[t] \,. \tag{9}$$

Here $\boldsymbol{u}^l[t]$ and $\boldsymbol{o}^l[t]$ denote the membrane potential and binary spike outputs *of all neurons* in $l$-th layer at time-step $t$, and $\boldsymbol{W}^l$ denotes the synaptic weights between layer $l-1$ and layer $l$. We assign different positive weights $\boldsymbol{\omega}^l[t] > \boldsymbol{0}$ to each time-step which is different from Deng et al. (2022) and $\hat{\boldsymbol{\gamma}}[t] = \boldsymbol{\omega}[t]\boldsymbol{\gamma}[t], \hat{\boldsymbol{\beta}}[t] = \boldsymbol{\omega}[t]\boldsymbol{\beta}[t]$. The weights $\boldsymbol{\omega}[t]$ and parameters $\boldsymbol{\gamma}^l[t], \boldsymbol{\beta}^l[t]$ are learnable, which are trained during the training process. For details, refer to Append. A and Append. B. Refer to Append. C for the learning rules to compute the gradients.

Computation of the temporal accumulated statistics is dynamically performed, in a moving averaging fashion, without the need to store batch data from all previous time-steps. This not only saves memory, but is also an important feature of our novel approach. For the algorithm details of the TAB method, please refer to algorithm 1 in the Appendix.

The rationale behind employing this accumulated spatial-temporal information in TAB comes from the sequential processing and temporal dependency characteristics intrinsic to spiking neurons. The TAB method utilizes the accumulated batch statistics $(\mu_{1:t}, \sigma_{1:t}^2)$ over an expanding window $[1, t]$. Fig. 2 illustrates an overview of four typical BN methods used in SNNs: default BN (Ioffe & Szegedy, 2015), BNTT (Kim & Panda, 2021), tdBN (Zheng et al., 2021), and TEBN (Duan et al., 2022). A comprehensive overview of statistics and parameters used by these methods is summarized in Table S1 in the Append. B.

As shown in Table S1, BNTT (Kim & Panda, 2021) considers BN statistics at each time-step individually and calculates different BN statistics $(\mu[t], \sigma^2[t])$ and BN parameters $(\gamma[t])$ at each time-step, which ignores the temporal dependencies of the input spikes. In contrast, tdBN (Zheng et al., 2021) computes the same overall BN statistics $(\mu_{1:T}, \sigma_{1:T}^2)$ and BN parameters $(\gamma, \beta)$ across all time-steps, but overlooking the temporal differences. Similarly, TEBN (Duan et al., 2022) employs the same overall BN statistics $(\mu_{1:T}, \sigma_{1:T}^2)$ as tdBN, but introduces distinct weight parameters $p[t]$ at each time-step to capture time-specific variations. However, both tdBN and TEBN, computing BN statistics over $T$ time-steps, *implicitly assume* access to data from all $T$ time-steps, that is, even if the current time-step is $t < T$, future information up to time-step $T$ can also be obtained, which is not true for the temporal accumulation of membrane potential nor the neural dynamics. As illustrated in Fig. 2, the input statistics of tdBN and TEBN consider the statistics of all the time-steps and all effective batches, while BNTT considers BN statistics at each time-step. Despite these differences, none of the existing methods have addressed the alignment with the membrane potential accumulation.

## 3.3 THEORETICAL CONNECTION BETWEEN TAB METHOD AND THE NEURAL DYNAMICS

TAB is tailored to capture the temporal dependencies of neurons as closely as possible by aligning with the neuron dynamics. To explore the theoretical connection between the TAB method and the neural dynamics, we need to delve into the LIF dynamics from the perspective of differential equations. In SNNs, each neuron maintains the dynamics of its membrane potential $U(t)$ over time,

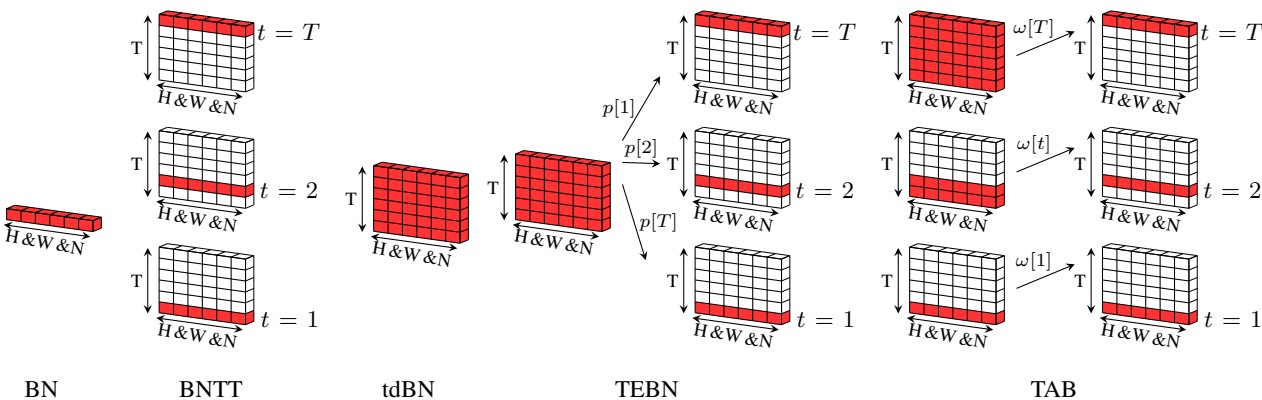

Figure 2: Comparison of different Batch Normalization methods with one given channel. In conventional BN, there is no time dimension. BNTT *independently* normalizes data at each time-step. The tdBN *jointly* normalizes data across all time-steps. TEBN shares a similar approach with tdBN but incorporates per-time-step scaling of the normalized data. In contrast, our TAB normalizes data using temporal accumulated statistics up to time-step $t$ and subsequently applies scaling.

by "integrating" the received input current $I(t)$ with a leakage term until a spike is triggered. This is described as a first-order linear differential equation (ODE),

$$\text{Neuron Dynamics as an ODE:} \quad \tau \frac{dU(t)}{dt} = -U(t) + RI(t), \quad U(t) < V_{th}, \tag{10}$$

where $I(t)$ represents the input current injected into the neuron at time $t$, and it is a function of $t$ (note that $I(t)$ is not a constant value). The closed-form solution of the LIF neuron dynamics (as an ODE) can be derived with analytical and theoretical methods. Additional details are available in Append. D.1 and Append. D.2.

**Lemma 1.** *The analytical closed-form solution for the first-order IVP (Initial Value Problem) of the LIF dynamics ODE is as follows (Gerstner et al., 2014),*

$$U(t) = \exp\left(-\frac{t}{\tau}\right)\left(\int_0^t \frac{R}{\tau}I(s)\exp\left(\frac{s}{\tau}\right)ds + U_0\right). \tag{11}$$

**Remark 1.** When the neuron initiates at the value $U_0$ with no further input, i.e., $I(t) = 0$, the closed-form solution of the ODE Eq. (11) shows that the membrane potential $U(t)$ will start at $U_0$ and exponentially decay with a time constant $\tau$, $U(t) = U_0\exp\left(-\frac{t}{\tau}\right)$. Consequently, we can determine the membrane potential ratio, often referred to as the leak factor, denoted by $\lambda$, as $\lambda = \frac{U(t+\Delta t)}{U(t)} = \frac{U_0\exp\left(-\frac{t+\Delta t}{\tau}\right)}{U_0\exp\left(-\frac{t}{\tau}\right)} = \exp\left(-\frac{\Delta t}{\tau}\right)$. This relationship enables us to formulate the discretization scheme as: $U[t+1] = \lambda U[t]$.

This remark provides insights into the behavior of the membrane potential in the absence of input and establishes the discretization principle used for LIF modeling.

**Lemma 2.** *Through applying integration by parts, we derive another equivalent form of the closed-form solution for the LIF dynamics, denoted as:*

$$U(t) = \overbrace{(U_0 - RI_0)\exp\left(-\frac{t}{\tau}\right)}^{\text{exponential decay term}} + \overbrace{RI(t)}^{\text{input current model}} - \underbrace{\int_0^t R\exp\left(\frac{s-t}{\tau}\right)dI(s)}_{\text{absent in the discrete LIF model}}. \tag{12}$$

*With the application of the Riemann–Stieltjes integral, the discretization version of the closed-form solution is represented as:*

$$U[t] = \overbrace{\lambda U[t-1]}^{(U_0 - RI_0)\exp\left(-\frac{t}{\tau}\right)} + \overbrace{X[t]}^{WO[t]=RI[t]} - \underbrace{\sum_{i=0}^{n} g_i X[s_i]}_{\text{TAB method}}. \tag{13}$$

In this formulation Eq. (13), the first exponential decay term, $\lambda U[t-1]$, captures the temporal dependency of the membrane potential from the preceding time-step. The second term, a simple current input model, $RI[t] = WO[t]$, incorporates spikes from the pre-connected neurons at the current time-step $[t]$. Significantly, the third term, representing the temporal accumulated input across all previous time-steps through a weighted sum of the input currents $X[s_i]$ with associated weights $g_i$, introduces a novel concept. Here $0 = s_0 < \cdots < s_i < \cdots < s_n = t$ denotes a partition of the time interval $[0, t]$ with a finite sequence of numbers. Refer to Append. D.3 for the details. Importantly, note that this accumulation mechanism of the inputs is a foundational component of the TAB method, providing a link that connects the TAB method and the neural dynamics.

**Remark 2.** The commonly used discrete LIF model in Eq. (2), as denoted by $U[t] = \lambda U[t-1] + X[t]$, is derived from the first two terms of the discretization version of the closed-form solution Eq. (13). The third term, representing the temporal accumulated input across all previous time-steps, however, is not incorporated into the discrete LIF models typically used in practice.

**Remark 3.** Note that the recursive application of the discrete LIF model, as denoted by $U[t] = \lambda U[t-1] + X[t]$, yields the temporal evolution of the membrane potential as $U[t] = \lambda^t U[0] + \sum_{s=1}^{t} \lambda^{t-s} X[s]$. This result shows the temporal dependency of the membrane potential accumulation in LIF neuron dynamics.

Recalling the TAB method introduced in Sect. 3.2, our TAB method normalizes data utilizing temporal accumulated batch statistics $(\mu_{1:t}, \sigma_{1:t}^2)$ across an expanding window $[1, t]$, where $\mu_{1:t}$ and $\sigma_{1:t}^2$ represent the temporal accumulated information up to time-step $[t]$. The utilization of the temporal accumulated batch statistics aligns well with the accumulation mechanism of the membrane potentialthrough Eq. (13). Consequently, it alleviates the temporal covariate shift issue which refers to the changes in the distribution of layer inputs resulting from updates of preceding layers and prior time-steps. The entire TAB method procedure and membrane updates can be linked through Eq. (13), derived by solving the LIF dynamics ODE. This equation naturally connects TAB batch normalization to neuron dynamics, as evident in Eq. (13).

Upon comparing the commonly used discrete LIF model in Eq. (2) with the discrete closed-form solution in Eq. (13), it shows that the TAB method reintroduces the accumulation term into the normalization procedure. This is achieved by using *temporal accumulated batch statistics* from time-step 1 to $t$. While the *temporal accumulated batch statistics* employed by the TAB method do not replicate the exact term in Eq. (13), but as an approximation. Thus, there exists no one-to-one functional mapping between the two. The adjustment within TAB method brings the discrete LIF model closer to its analytical closed-form counterpart, thus, TAB can work well in addressing the temporal covariate shift issue. This establishes a natural connection between neuron dynamics and batch normalization.

## 4 EXPERIMENTS

In this section, we conduct extensive experiments on large-scale static and neuromorphic datasets, CIFAR-10 (Krizhevsky et al., 2009), CIFAR100 (Krizhevsky et al., 2009), and DVS-CIFAR10 (Li et al., 2017), to verify the effectiveness of our proposed TAB method. We utilize the VGG network architecture and ResNet architecture. Firstly, we perform a comparative analysis of our TAB method with other BN methods in the context of SNNs. Further, we compare our TAB method with other state-of-the-art approaches. For implementation details, refer to Append. E.

### 4.1 COMPARISON WITH OTHER BN METHODS

We conduct our evaluation by comparing the performance of the proposed TAB method and other batch normalization methods in the context of SNNs. To ensure fairness in our comparisons, we do not employ advanced data augmentation techniques like cutout (DeVries & Taylor, 2017). Table 1 provides a comprehensive overview of the the test accuracy on both traditional static dataset CIFAR-10, CIFAR-100 and neuromorphic dataset DVS-CIFAR10. On the CIFAR-10 dataset, our TAB method demonstrates remarkable performance improvement, achieving a top-1 accuracy of $94.73\%$ with the ResNet-19 network using only 2 time-steps. Notably, this surpasses the performance of TEBN using 6 time-steps. Furthermore, when using the same network architecture, TAB consistently outperforms other BN methods, even with fewer time-steps $T$. This pattern holds true for other

Table 1: Comparison between the proposed TAB method and other BN methods in SNNs.

| Dataset | Model | Method | Architecture | Time-steps | Accuracy (%) |
|---------|-------|--------|--------------|------------|--------------|
| CIFAR-10 | SPIKE-NORM (Sengupta et al., 2019) | ANN-to-SNN | VGG-16 | 2500 | 91.55 |
| | NeuNorm (Wu et al., 2019) | Surrogate Gradient | CIFARNet | 12 | 90.53 |
| | BNTT (Kim & Panda, 2021) | Surrogate Gradient | VGG-9 | 20 | 90.30 |
| | tdBN (Zheng et al., 2021) | Surrogate Gradient | ResNet-19 | 6 / 4 / 2 | 93.16 / 92.92 / 92.34 |
| | TEBN (Duan et al., 2022) | Surrogate Gradient | VGG-9 | 4 | 92.81 |
| | | | ResNet-19 | 6 / 4 / 2 | 94.71 / 94.70 / 94.57 |
| | **TAB (Ours)** | Surrogate Gradient | VGG-9 | 4 | **93.41** |
| | | | ResNet-19 | 6 / 4 / 2 | **94.81 / 94.76 / 94.73** |
| CIFAR-100 | SPIKE-NORM (Sengupta et al., 2019) | ANN-to-SNN | VGG-16 | 2500 | 70.90 |
| | BNTT (Kim & Panda, 2021) | Surrogate Gradient | VGG-11 | 50 | 66.60 |
| | TEBN (Duan et al., 2022) | Surrogate Gradient | VGG-11 | 4 | 74.37 |
| | TEBN (Duan et al., 2022) | Surrogate Gradient | ResNet-19 | 6 / 4 / 2 | 76.41 / 76.13 / 75.86 |
| | **TAB (Ours)** | Surrogate Gradient | VGG-11 | 4 | **75.89** |
| | | | ResNet-19 | 6 / 4 / 2 | **76.82 / 76.81 / 76.31** |
| DVS-CIFAR10 | NeuNorm (Wu et al., 2019) | Surrogate Gradient | 7-layer CNN | 40 | 60.50 |
| | BNTT (Kim & Panda, 2021) | Surrogate Gradient | 7-layer CNN | 20 | 63.2 |
| | tdBN (Zheng et al., 2021) | Surrogate Gradient | ResNet-19 | 10 | 67.8 |
| | TEBN (Duan et al., 2022) | Surrogate Gradient | 7-layer CNN | 10 | 75.10 |
| | **TAB (Ours)** | Surrogate Gradient | 7-layer CNN | 4 | **76.7** |
| ImageNet | SlipReLU (Jiang et al., 2023) | ANN-to-SNN | ResNet-34 | 32 | 66.61 |
| | tdBN (Zheng et al., 2021) | Surrogate Gradient | ResNet-34 | 6 | 63.72 |
| | TEBN (Duan et al., 2022) | Surrogate Gradient | ResNet-34 | 4 | 64.29 |
| | **TAB (Ours)** | Surrogate Gradient | ResNet-34 | 4 | **67.78** |
| | | | ResNet-34 | 2 | **65.94** |

datasets as well. For instance, on the DVS-CIFAR10 dataset, our TAB method achieves 1.6% better performance (76.7% v.s. 75.10%) while utilizing fewer time-steps (4 v.s. 10) than TEBN. Similarly, on CIFAR-100, our method exhibits a 0.55% increase in accuracy (76.31% v.s. 75.86%) compared to TEBN when both use 2 time-steps. All the accuracy values for other methods reported in the table are drawn from the existing literature.

## 4.2 COMPARISON ON LARGE-SCALE IMAGENET DATASET

In this section, we investigate the effectiveness of our TAB method on the ImageNet dataset, renowned for its extensive collection of more than 1.25 million training images and $50,000$ test images (Deng et al., 2009). The training set of ImageNet offers $1,280$ training samples for each label, and we apply standard preprocessing and augmentation techniques (He et al., 2016) to the training data. Test data is centered and cropped to dimensions of $224 \times 224$. The evaluation employs the ResNet-34 architecture, a widely recognized model. The network is trained using the AdamW optimizer with an initial learning rate of 0.00002 and a weight decay of 0.02. Training occurs on an NVIDIA RTX A6000 with 4 GPUs, each handling a batch size of 24. To ensure unbiased statistics, we follow Zheng et al. (2021) and synchronize batch mean and variance across devices.

The results, presented in Tables Table 1 and Table S4, reveal the efficacy of our TAB method. Notably, even with a modest training duration of 80 epochs for $T = 4$, the TAB method exhibits a 3.29% improvement on ResNet-34 over TEBN at $T = 4$ (TAB with 67.78% vs. TEBN 64.29%). Impressively, with only 2 time-steps ($T = 2$), our TAB method achieves an accuracy of 65.94% on ImageNet, showcasing its promising performance.

## 4.3 COMPARISON WITH THE STATE-OF-THE-ART APPROACHES

In this section, we present a comprehensive comparison of our TAB method with other state-of-the-art learning methods for SNNs using CIFAR-10 as the benchmark dataset, as illustrated in Table 2.

On the VGG-11 architecture, our TAB method achieves an impressive accuracy of 94.73% while utilizing 4 time-steps, outperforming all the ANN-to-SNN conversion and hybrid training methods that require more time-steps. Besides, we follow TEBN (Duan et al., 2022) and adopt the cutout augmentation (DeVries & Taylor, 2017) on static datasets denoted by "*" in the table. Compared to other surrogate gradient methods, our TAB method consistently performs better. On ResNet-19, our TAB method achieves an accuracy of 96.09% with 6 time-steps, which is better than Dspike (94.25%), TET (94.5%), TEBN (95.6%) while using the same number of time-steps. Even when using only 2 time-steps $T = 2$, our TAB method on ResNet-19 achieves a higher accuracy than TEBN (Duan et al., 2022) which utilizes 6 time-steps. We contribute this elevated performance to the better representation capability of TAB, achieved by its alignment with the neuron dynamics, thereby bridging the gap between the discrete LIF model and the underlying neuron dynamics and making the two closer. For clarity, all reported accuracy values for other methods in the tables are sourced from the literature. Further experimental results on CIFAR-100 and DVS-CIFAR10 datasets are detailed in Table S3 from Append. E. For a comprehensive comparison with state-of-the-art (SOTA) methods on ImageNet, please consult Table S4 provided in Append. E.5.

Table 2: Comparison between the proposed TAB and other state-of-the-art approaches on CIFAR-10.

| Model | Method | Architecture | Time-steps | Accuracy (%) |
|---|---|---|---|---|
| RMP (Han et al., 2020) | ANN-to-SNN | ResNet-20 | 2048 | 91.36 |
| RTS (Deng & Gu, 2021) | ANN-to-SNN | ResNet-20 | 128 | 93.56 |
| QCFS (Bu et al., 2022) | ANN-to-SNN | ResNet-20 | 16 | 91.62 |
| PTL (Wu et al., 2021b) | ANN-to-SNN | VGG-11 | 16 | 91.24 |
| HC (Rathi et al., 2020) | Hybrid Training | VGG-11 | 2500 | 92.94 |
| TC (Zhou et al., 2021) | Time-based Gradient | VGG-16 | - | 92.68 |
| TSSL-BP (Zhang & Li, 2020) | Time-based Gradient | 7-layer CNN | 5 | 91.41 |
| Dspike (Li et al., 2021b) | Surrogate Gradient | ResNet-18* | 6 / 4 / 2 | 94.25 / 93.66 / 93.13 |
| TET (Deng et al., 2022) | Surrogate Gradient | ResNet-19* | 6 / 4 / 2 | 94.50 / 94.44 / 94.16 |
| TEBN (Duan et al., 2022) | Surrogate Gradient | VGG-11 | 4 | 93.96 |
| TEBN (Duan et al., 2022) | Surrogate Gradient | ResNet-19* | 6 / 4 / 2 | 95.60 / 95.58 / 95.45 |
| **TAB (Ours)** | Surrogate Gradient | VGG-11 | 4 | **94.73** |
| | | ResNet-19* | 6 / 4 / 2 | **96.09 / 95.94 / 95.62** |

## 5 Conclusion

Directly training SNNs is extremely challenging, even when adopting BN techniques to enable more stable training. The presence of the Temporal Covariate Shift (TCS) phenomenon, coupled with the intrinsic temporal dependency of neuron dynamics, further compounds these challenges for directly training SNNs. To tackle this, we have introduced TAB (Temporal Accumulated Batch Normalization), a novel SNN batch normalization approach. TAB closely aligns with the neuron dynamics, normalizing data using temporal accumulated statistics, effectively capturing historical temporal dependencies similar to that of the accumulation process of the membrane potential in the LIF neuron model. Neuron dynamics refer to the changes in the membrane potential of a neuron over time as it integrates input signals and generates spikes. The alignment with the neuron dynamics means that the TAB method is tailored to mimic or capture the behavior of neurons as closely as possible. It aims to normalize the data in a manner that is coherent with the temporal dependencies and accumulation of information that occur within neurons as they process input signals. This alignment ensures that TAB's normalization process corresponds well with the way neurons naturally operate in SNNs, thereby leading to improved training and performance by addressing the temporal covariate shift problem.

## Acknowledgements

This work is part of the research project ("Energy-based probing for Spiking Neural Networks", Contract No. TII/ARRC/2073/2021) in collaboration between Technology Innovation Institute (TII, Abu Dhabi) and Mohamed bin Zayed University of Artificial Intelligence (MBZUAI, Abu Dhabi).

## REPRODUCIBILITY STATEMENT

The experiments and results presented in this research are reproducible, with all code, data, and detailed methodologies available in the supplementary materials. The codebase has been documented extensively, ensuring clarity and ease of implementation for future researchers. The datasets used in this study, including CIFAR-10, CIFAR-100, and DVS-CIFAR10, are publicly accessible, and we provide precise instructions on data preprocessing and augmentation procedures. Additionally, the hardware and software specifications utilized for conducting experiments are thoroughly documented, enabling researchers to replicate our results under similar computational environments. We are committed to supporting the scientific community's efforts in validating and building upon our work, thus promoting transparency and trustworthiness in the field of spiking neural networks and batch normalization techniques.

## ETHICS STATEMENT

This research strictly adheres to ethical standards and guidelines governing scientific inquiry. All experiments involving living subjects or animals were not a part of this study, eliminating any ethical concerns in that regard. In terms of data usage, we employed publicly available datasets, ensuring no breach of privacy or data protection regulations. In terms of research conduct, this study promotes openness and transparency by making all code, data, and methodologies accessible to the wider scientific community. We also acknowledge and properly cite prior work, respecting intellectual property rights and academic integrity. Furthermore, this research focuses on improving the efficiency and effectiveness of spiking neural networks, which could potentially contribute to more energy-efficient AI applications. We are committed to upholding the highest ethical standards in research and encourage responsible and transparent scientific practices within the field.

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

# Appendices

SUPPLEMENTARY MATERIAL FOR "TAB: TEMPORAL ACCUMULATED BATCH NORMALIZATION IN SPIKING NEURAL NETWORKS"

NOTATIONS

Compared to ANNs, SNNs utilize binary activations, or spikes, in each layer. To compensate for this limited representation capacity of the binary spiking activation, the time dimension, also known as latency $T$, is introduced in SNNs. For the forward pass in SNNs, inputs are presented as streams of events and the forward pass is repeated for $T$ time-steps to produce the final result.

To make it clear, we refer to Eq. (1) as the LIF neuron dynamics which are described as differential equations, and refer to Eq. (2) as the discrete LIF neuron model.

Let $x[t]$ be the spike representation of the traditional features $x$ at time-step $t$, denoted as $x[t] = SpikeRep(x, t)$. Here, let $(x, y)$ be samples from a traditional dataset $\mathcal{D}$ with $x$ representing the image and $y$ the label. Note that $x$ and $x[t]$ share the same dimensionality, w.l.o.g., let $x \in \mathbb{R}^{C \times H \times W}$ have $C$ as the channel dimension, $(H, W)$ as the spatial dimension, so $x[t] \in \mathbb{R}^{C \times H \times W}$. Let $\mathfrak{D}_t$ represent the set of spiking representations at time-step $t$, which can be expressed as:

$$\mathfrak{D}_t = \{(x[t], y), \text{ with } x[t] = SpikeRep(x, t) \text{ and } \forall \, (x, y) \in \mathcal{D}\} \, .$$

Due to the temporal processing nature of SNNs, the training dataset $\mathfrak{D}$ consists of sets of spiking representations from all time-steps, which satisfies $\mathfrak{D} = \mathfrak{D}_1 \cup \cdots \cup \mathfrak{D}_T$, and any two are disjoint with others, $\mathfrak{D}_t \cap \mathfrak{D}_s = \emptyset$ if $t \neq s$.

---

**Algorithm 1:** The Algorithm of TAB Method

---

**Data:** The mini-batch data at current time-step $t$, $\{B_t\} = \{x_i[t]\}_{i=1}^b$.
**Input:** Parameters to be learned: $\omega[t], \gamma[t], \beta[t]$, where weight $\omega[t] > 0$.
**Output:** $\hat{x}_i[t] = \text{TAB}(x_i[t])$

1 **for** $t = 1, \cdots, T$ **do**

2      Calculate Mean/Variance for mini-batch $B_t$ at current time-step $t$
        $\mu[t] = \frac{1}{b} \sum_{i=1}^b x_i[t] \, , \quad \sigma^2[t] = \frac{1}{b} \sum_{i=1}^b (x_i[t] - \mu[t])^2$

3      Calculate Temporal Accumulated Mean/Variance

$$\mu_{1:t} = \frac{t-1}{t} \mu_{1:(t-1)} + \frac{1}{t} \mu[t] \, , \quad \sigma^2_{1:t} = \frac{t-1}{t} \sigma^2_{1:(t-1)} + \frac{1}{t} \sigma^2[t]$$

4      The TAB Method

$$z_i[t] \leftarrow \gamma[t] \frac{x_i[t] - \mu_{1:t}}{\sqrt{\sigma^2_{1:t} + \epsilon}} + \beta[t] \, , \text{ for } i = 1, \cdots, b$$

$$\hat{x}_i[t] = \omega[t] z_i[t] = \text{TAB}(x_i[t]) \, , \quad \omega[t] > 0 \, .$$

5      Outputs: $\hat{x}[t] = \{\hat{x}_{1 \ldots b}[t]\}$

---

## A    BN OF CONVOLUTIONAL LAYERS

For convolutional layers, we additionally want the normalization obey the convolutional property, that is, different elements of the same feature map (or the same channel), at different locations, are normalized in the same way. Specifically, let $x^l[t]$ denote the input variable to the layer $l$ at time-step $t$, where $x^l[t]$ is a tensor with four dimensions $N \times C \times H \times W$. Denote $x^l[t] \in \mathbb{R}^{N \times C \times H \times W}$ the corresponding batch input data to the layer $l$ at time-step $t$. We let $B_t$ be the set of all values in a feature map across both the elements of a mini-batch and spatial locations. So for a mini-batch of size $N$ and feature maps of size $H \times W$, we use the effective mini-batch of size $b = |B_t| = NHW$.

The mini-batch data can be written as $\boldsymbol{x}^l[t] = (\boldsymbol{x}_1^l[t], \cdots, \boldsymbol{x}_C^l[t])$, where each $\boldsymbol{x}_c^l[t] \in \mathbb{R}^{N \times H \times W}$ denotes the input to the layer $l$ at time-step $t$ for $c$-th channel (or $c$-th feature map). Since the normalization is applied *independently* to each dimension over the samples in a mini-batch $\boldsymbol{x}_c^l[t]$, let us focus on the mini-batch data of a particular dimension $\boldsymbol{x}_c^l[t]$ and omit $c$ for clarity. Consider a spatial-temporal mini-batch $B_t = \{\boldsymbol{x}^l[t]\} = \boldsymbol{x}_{1 \dots b}[t]$ of size $b$ at each time-step $t$.

# B  DETAILS OF TAB IN SPIKING NEURAL NETWORKS

Our proposed TAB normalizes data along the channel dimension before passing it through the activation function, then this normalized data is fed as an input to the activation function. Let $\boldsymbol{x}^l[t]$ denote the input variable to layer $l$ at time-step $t$. Without loss of generality, we assume that the layer input $\boldsymbol{x}^l[t]$ has $d$-dimensions, which can be written as $\boldsymbol{x}^l[t] = (\boldsymbol{x}^{l(1)}[t], \cdots, \boldsymbol{x}^{l(d)}[t])$. Here $\boldsymbol{x}^{l(k)}[t]$ represents the $k$-th dimensional input to the $l$-th layer at time-step $t$. Since the normalization is performed independently on each dimension $\boldsymbol{x}^{l(k)}[t]$, we can focus on a specific dimension denoted as $\boldsymbol{x}[t]$ and omit the $l(k)$ notation for clarity.

We investigate Spiking Convolutional Neural Networks (SCNNs), which are Spiking Neural Networks that incorporate convolutional layers and can process both spatial and temporal information. Our main focus is primarily on applying Batch Normalization for Spiking Convolutional Neural Networks. In SNNs, there is an additional temporal dimension, indexed by time-step $[t]$. Thus, at the current time-step $[t]$, all the ***accumulated spatial-temporal mini-batches***, denoted as $B_{1:t} = \{B_s\}_{s=1}^t$, are available, where the temporal information is accumulated over $t$ time-steps. Here, $B_s = \{\boldsymbol{x}_{b,c,h,w}[s]\}$ represents a ***spatial-temporal mini-batch*** at time-step $s$ with batch index $b$, channel index $c$, height index $h$, and width index $w$. For notation simplification, we can put all the batch dimension $N$ and all spatial locations $H, W$ into one dimension with $b = NHW$ and indexed by $i$, and we also omit the channel index $c$ for a given channel. In particular, let $x_i[t]$ represent the $i$-th sample at time-step $t$ in the spatial-temporal mini-batch $B_t$.

Let $\boldsymbol{o}^l[t]$ denote the spiking outputs of all neurons in $l$-th layer at time-step $t$, and $\boldsymbol{W}^l$ denote the synaptic weights between layer $l-1$ and layer $l$. The pre-synaptic inputs to the LIF neurons, denoted as $\boldsymbol{x}^l[t]$ can be expressed as

$$\boldsymbol{x}^l[t] = \boldsymbol{W}^l * \boldsymbol{o}^{l-1}[t] \,.$$

In Spiking Convolutional Neural Networks, both the spikes and the pre-synaptic inputs at time-step $t$ are 3-dimensional tensors. Specifically, we refer to $\boldsymbol{o}^{l-1}[t] := \boldsymbol{o}_{b,c,h,w}^{l-1}[t] \in \mathbb{R}^{N \times C \times H \times W}$ as the spikes tensor and $\boldsymbol{x}^l[t] := \boldsymbol{x}_{b,c,h,w}^l[t] \in \mathbb{R}^{N \times C \times H \times W}$ as the pre-synaptic inputs tensor in a mini-batch with batch size $N$. Here, the indices $b, c, h, w$ provide a precise description of the dimensions, representing the batch axis $N$, the channel axis $C$, and the two spatial dimensions $H$ and $W$, respectively. In the case of input images to the neural network, the channels correspond to the RGB channels.

The ***spatial-temporal mini-batch*** $B_t$ at time-step $t$, with batch size $N$, is defined as the set of all values in a feature map across both the elements of a mini-batch and spatial locations. For a mini-batch of size $N$ and feature maps of size $H \times W$, the effective mini-batch of size is $b = |B_t| = NHW$. We learn a pair of parameters, $\gamma^{(k)}$ and $\beta^{(k)}$, per feature map, where $k = 1, \cdots, C$, rather than per activation.

**TAB with layer index.**  The TAB Layer normalizes the pre-synaptic inputs $\boldsymbol{x}^l[t]$ in the $l$-th layer of the spiking neuron networks at time-step $t$, and it can be formulated as

$$\boldsymbol{x}^l[t] = \boldsymbol{W}^l \boldsymbol{o}^{l-1}[t] \,,$$
$$\boldsymbol{u}^l[t] = \lambda \boldsymbol{u}^l[t-1](1 - \boldsymbol{o}^l[t-1]) + \boldsymbol{y}^l[t] \,,$$
$$\text{where} \quad \boldsymbol{y}^l[t] = \text{TAB}(\boldsymbol{x}^l[t]) = \boldsymbol{\omega}^l[t]\hat{\boldsymbol{x}}^l[t] \,, \quad \boldsymbol{\omega}^l[t] \geqslant \boldsymbol{0}$$
$$\hat{\boldsymbol{x}}^l[t] = \boldsymbol{\gamma}^l[t]\frac{\boldsymbol{x}^l[t] - \boldsymbol{\mu}_{1:t}^l}{\sqrt{\boldsymbol{V}_{1:t}^l + \epsilon}} + \boldsymbol{\beta}^l[t] \,.$$

Here $\boldsymbol{u}^l[t]$ and $\boldsymbol{o}^l[t]$ represent the membrane potential and binary spike outputs, respectively, of ***all neurons*** in $l$-th layer at time-step $t$. The variable $\boldsymbol{x}^l[t]$ represents the input variable to layer $l$, and

$W^l$ denotes the synaptic weights between layer $l-1$ and layer $l$. Each TAB Layer incorporates time-dependent learnable parameters, namely $\boldsymbol{\omega}^l[t], \boldsymbol{\gamma}^l[t], \boldsymbol{\beta}^l[t]$, for each layer. These parameters are $C$-channel tensors defined as $\boldsymbol{\omega}^l[t] = (\omega_1^l[t], \cdots, \omega_C^l[t]), \boldsymbol{\gamma}^l[t] = (\gamma_1^l[t], \cdots, \gamma_C^l[t]), \boldsymbol{\beta}^l[t] = (\beta_1^l[t], \cdots, \beta_C^l[t])$. The learnable parameters $\boldsymbol{\omega}^l[t], \boldsymbol{\gamma}^l[t], \boldsymbol{\beta}^l[t]$ $(t = 1, \cdots, T, l = 1 \cdots, L)$ are trained during the training process.

Here the accumulated mean $\boldsymbol{\mu}_{1:t}^l$ and the accumulated variance $\boldsymbol{V}_{1:t}^l = (\boldsymbol{\sigma}_{1:t}^l)^2$ are computed as follows,

$$\boldsymbol{\mu}_{1:t}^l = \frac{1}{t}\sum_{s=1}^{t}\boldsymbol{\mu}^l[s]\,, \quad \boldsymbol{V}_{1:t}^l = (\boldsymbol{\sigma}_{1:t}^l)^2 = \frac{1}{t}\sum_{s=1}^{t}(\boldsymbol{\sigma}^l[s])^2\,,$$

where

$$\boldsymbol{\mu}^l[s] = \frac{1}{b}\sum_{i=1}^{b}\boldsymbol{x}_i^l[s]\,, \quad (\boldsymbol{\sigma}^l[s])^2 = \frac{1}{b}\sum_{i=1}^{b}(\boldsymbol{x}_i^l[s] - \boldsymbol{\mu}^l[s])^2.$$

If we expand the channel dimension, we will have the following. Here the accumulated mean $\boldsymbol{\mu}_{1:t}^l = (\boldsymbol{\mu}_{1,1:t}^l, \cdots, \boldsymbol{\mu}_{C,1:t}^l)$ where each $\boldsymbol{\mu}_{c,1:t}^l$ has the same definition on each channel, and the accumulated variance $\boldsymbol{V}_{1:t}^l = (\boldsymbol{V}_{1,1:t}^l, \cdots, \boldsymbol{V}_{C,1:t}^l)$ is defined analogously. Specifically,

$$\boldsymbol{\mu}_{1:t}^l = \frac{1}{t}\sum_{s=1}^{t}\boldsymbol{\mu}^l[s]\,, \quad \boldsymbol{V}_{1:t}^l = (\boldsymbol{\sigma}_{1:t}^l)^2 \approx \frac{1}{t}\sum_{s=1}^{t}(\boldsymbol{\sigma}^l[s])^2\,,$$

where notations $\boldsymbol{\mu}^l[s] = (\boldsymbol{\mu}_1^l[s], \cdots, \boldsymbol{\mu}_C^l[s])$ and $(\boldsymbol{\sigma}^l[s])^2 = ((\boldsymbol{\sigma}_1^l[s])^2, \cdots, (\boldsymbol{\sigma}_C^l[s])^2)$, and Specially,

$$\boldsymbol{\mu}_c^l[s] = \frac{1}{NHW}\sum_{b,h,w}\boldsymbol{x}_{b,c,h,w}^l[s]\,, \quad (\boldsymbol{\sigma}_c^l[s])^2 = \frac{1}{NHW}\sum_{b,h,w}(\boldsymbol{x}_{b,c,h,w}^l[s] - \boldsymbol{\mu}_c^l[s])^2.$$

**TAB without layer index.** As here we normalize all the neurons in $l$-th layer at time-step $t$, we can omit the layer index $l$. In the batch normalization process, we will *independently* normalize each scalar feature along the channel dimension, by making it have zero mean and the variance of 1 over a *mini-batch*. In our TAB method, the pre-synaptic inputs $\boldsymbol{x}[t]$ will be normalized and both the input and output of a TAB layer are four-dimensional tensors. Our proposed TAB applies the same normalization for all activations in a given channel (along the channel dimension),

$$\boldsymbol{y}_{b,c,h,w}[t] \leftarrow \omega_c[t]\left(\gamma_c[t]\frac{\boldsymbol{x}_{b,c,h,w}[t] - \mu_{c,1:t}}{\sqrt{\sigma_{c,1:t}^2 + \epsilon}} + \beta_c[t]\right)\,. \tag{S.1}$$

The details of the TAB

$$\boldsymbol{y}_{b,c,h,w}[t] = \text{TAB}(\boldsymbol{x}_{b,c,h,w}[t]) = \omega_c[t]\hat{\boldsymbol{x}}_{b,c,h,w}[t]\,, \quad \omega_c[t] > 0\,, \tag{S.2}$$

where $\quad \hat{\boldsymbol{x}}_{b,c,h,w}[t] = \gamma_c[t]\dfrac{\boldsymbol{x}_{b,c,h,w}[t] - \mu_{c,1:t}}{\sqrt{\sigma_{c,1:t}^2 + \epsilon}} + \beta_c[t]\,, \tag{S.3}$

$$\mu_{c,1:t} = \frac{1}{t}\sum_{s=1}^{t}\mu_c[s]\,, \quad \mu_c[s] = \frac{1}{NHW}\sum_{b,h,w}\boldsymbol{x}_{b,c,h,w}[s]\,, \tag{S.4}$$

$$\sigma_{c,1:t}^2 = \frac{1}{t}\sum_{s=1}^{t}(\sigma_c[s])^2\,, \quad (\sigma_c[s])^2 = \frac{1}{NHW}\sum_{b,h,w}(\boldsymbol{x}_{b,c,h,w}[s] - \mu_{c,1:t})^2\,. \tag{S.5}$$

Here, TAB subtracts the **accumulated mean** activation $\mu_{c,1:t}$ from all input activations in channel $c$, where $B_{1:t}$ contains all activations in channel $c$ over the accumulated time-step from 1 to $t$ across all features $b$ in the entire mini-batch and all spatial $h, w$ locations. Subsequently, TAB divides the centered activation by the **accumulated standard deviation** $\sigma_{c,1:t}^2$ (plus $\epsilon$ for numerical stability) which is calculated analogously. Normalization is followed by a channel-wise affine transformation and scaling parametrized through $\gamma_c[t], \beta_c[t], \omega_c[t](\omega_c[t] > 0)$ at different time-step $t$, which are

learned during training. During testing, running averages of the accumulated mean and accumulated variances are used. Refer to algorithm 1 for the algorithm of our TAB method.

The reason for considering all the accumulated spatial-temporal mini-batch over an expanding window $[1, t]$ is enabled by the nature of the spiking neurons to process both spatial and temporal information. In SNNs, as the synaptic currents are fed into spiking neurons *sequentially*, the membrane potential $\boldsymbol{u}_i(t)$ accumulates over time $t$. The temporal information in SNNs is accumulated over multiple time-steps, allowing the neurons to consider the input data in context of the past time-steps as well as the current time-step. Therefore, we expect it to be beneficial for batch normalization to consider the temporal accumulated information over an expanding window $[1, t]$.

Table S1: Statistics and parameters of pre-synaptic inputs with different BN methods.

| | $t = 1$ | $t$ | $\cdots$ | $t = T$ |
|---|---|---|---|---|
| BNTT (Kim & Panda, 2021) | $\mu[1], \sigma^2[1]$ | $\mu[t], \sigma^2[t]$ | $\cdots$ | $\mu[T], \sigma^2[T]$ |
| | $\gamma[1]$ | $\gamma[t]$ | $\cdots$ | $\gamma[T]$ |
| tdBN (Zheng et al., 2021) | $\mu_{\text{total}} = \mu_{1:T}, \sigma^2_{\text{total}} = \sigma^2_{1:T}$ | | | |
| | $\gamma, \beta$ | | | |
| TEBN (Duan et al., 2022) | $\mu_{\text{total}} = \mu_{1:T}, \sigma^2_{\text{total}} = \sigma^2_{1:T}$ | | | |
| | $\hat{\gamma}[1], \hat{\beta}[1]$ | $\hat{\gamma}[t], \hat{\beta}[t]$ | $\cdots$ | $\hat{\gamma}[T], \hat{\beta}[T]$ |
| **TAB (ours)** | $\mu[1], \sigma^2[1]$ | $\mu_{1:t}, \sigma^2_{1:t}$ | $\cdots$ | $\mu_{1:t}, \sigma^2_{1:t}$ |
| | $\omega[1]\gamma[1], \omega[1]\beta[1]$ | $\omega[t]\gamma[t], \omega[t]\beta[t]$ | $\cdots$ | $\omega[T]\gamma[T], \omega[T]\beta[T]$ |

## C  DETAILED LEARNING RULES OF TAB IN SNNs

When applying TAB in SNNs, we derive the detailed learning rules by computing the gradients of the loss function $\mathcal{L}$ with respect to the weights $\partial W_{ij}^l$ and with respect to the parameter $\partial \omega^l[t]$. Following the previous work (Kim & Panda, 2021; Zheng et al., 2021; Duan et al., 2022), we compute the gradient by unfolding the network along the time dimension and get,

$$\frac{\partial \mathcal{L}}{\partial \boldsymbol{u}_i^l[t]} = \frac{\partial \mathcal{L}}{\partial \boldsymbol{o}_i^l[t]} \frac{\partial \boldsymbol{o}_i^l[t]}{\partial \boldsymbol{u}_i^l[t]} + \frac{\partial \mathcal{L}}{\partial \boldsymbol{u}_i^l[t+1]} \frac{\partial \boldsymbol{u}_i^l[t+1]}{\partial \boldsymbol{u}_i^l[t]}$$

$$\frac{\partial \mathcal{L}}{\partial W_{ij}^l} = \frac{\partial \mathcal{L}}{\partial \boldsymbol{x}_i^{l-1}[t]} \boldsymbol{o}_j^{l-1}[t] = \frac{\partial \mathcal{L}}{\partial \boldsymbol{u}_i^l[t]} \frac{\partial \hat{\boldsymbol{x}}_i^{l-1}[t]}{\partial \boldsymbol{x}_i^{l-1}[t]} \boldsymbol{o}_j^{l-1}[t]$$

where $\boldsymbol{u}_i^l[t]$ and $\boldsymbol{o}_i^l[t]$ denote the membrane potential and output spikes of the $i$-th neuron at time-step $t$ in layer $l$, $\boldsymbol{x}_i^l[t]$ and $\hat{\boldsymbol{x}}_i^l[t]$ are the input and output of the TAB layer. As the derivative of spikes with respect to the membrane potential $\frac{\partial \boldsymbol{o}_i^l[t]}{\partial \boldsymbol{u}_i^l[t]}$ is non-differentiable due to the Heaviside step function, we use the surrogate gradient methods (Neftci et al., 2019; Wu et al., 2018; Eshraghian et al., 2021) to address this problem.

The weight parameter $\boldsymbol{\omega}^l[t]$ allows the TAB layer to emphasize the temporal dynamics along the temporal dimension. The gradient of the loss function $\mathcal{L}$ with respect to the weight parameter $\partial \boldsymbol{\omega}^l[t]$,

$$\frac{\partial \mathcal{L}}{\partial \boldsymbol{\omega}^l[t]} = \sum_i \frac{\partial \mathcal{L}}{\partial \boldsymbol{u}_i^l[t]} \frac{\partial \boldsymbol{u}_i^l[t]}{\partial \boldsymbol{\omega}^l[t]} = \sum_i \left[ \frac{\partial \mathcal{L}}{\partial \boldsymbol{u}_i^l[t]} \left( \gamma_i^l[t] \frac{\boldsymbol{x}_i^l[t] - \boldsymbol{\mu}_{1:t}^l}{\sqrt{(\boldsymbol{\sigma}_{1:t}^l)_i + \epsilon}} + \boldsymbol{\beta}_i^l[t] \right) \right] ,$$

where $\boldsymbol{\mu}_{1:t}^l$ and $(\boldsymbol{\sigma}_{1:t}^l)_i$ denote the mean and variance of the $i$-th neuron from samples from time-step 1 to $t$, $\boldsymbol{\gamma}_i^l[t]$ and $\boldsymbol{\beta}_i^l[t]$ are the $i$-th parameters of TAB at time-step $t$.

## D  THEORETICAL POOF

In this section, we provide the theoretical proof for **Lemma 1** and **Lemma 2** in the main paper. We first give a short introduction to the first-order differential equations and the Initial Value Problem

(IVP). Then we derive the analytical closed-form solution of the first-order differential equation underlying the LIF neuron dynamics and give detailed proof for the main lemma.

### D.1 FIRST-ORDER DIFFERENTIAL EQUATIONS

We start by considering equations in which only the first derivative of the function appears.

**Definition 1** (First-Order Differential Equation). A first-order differential equation is an equation of the form $F(t, y, y') = 0$. A solution of a first-order differential equation is a function $f(t)$ that makes $F(t, f(t), f'(t)) = 0$ for every value of $t$. $\square$

The term "first-order" means that the first derivative of $y$ appears, but no higher order derivatives do. Note that the derivative $y'$ will *explicitly* appear in the equation.

**Definition 2** (First-Order Initial Value Problem). A first-order initial value problem is a system of equations of the form $F(t, y, y') = 0$, $y(t_0) = y_0$. Here $t_0$ is a fixed time and $y_0$ is a number. A solution of an initial value problem is a solution $f(t)$ of the differential equation that also satisfies the initial condition $f(t_0) = y_0$. $\square$

The general first-order equation is rather too general. We focus on specific kinds of first-order differential equations. For example, equations of the form $y' = \phi(t, y)$ where $\phi$ is a function of the two variables $t$ and $y$. Under reasonable conditions on $\phi$, such an equation has a solution and the corresponding initial value problem has a unique solution.

**Definition 3** (First-Order Linear Differential Equation). A first-order differential equation is ***linear*** when it can be written as

$$y' + P(t)y = Q(t) \tag{S.6}$$

Where $P(t)$ and $Q(t)$ are functions of $t$. $\square$

We now need to derive the general solution of a linear first-order differential equation. We start by revisiting two lemmas of first-order differential equation (MacCluer et al., 2019).

**Lemma S.1.** *(Homogeneous Solution (MacCluer et al., 2019).) If $P(t)$ is continuous on $(a, b)$, then the general solution of the homogeneous equation*

$$y' + P(t)y = 0 \, ,$$

*on $(a, b)$ is*

$$y = ce^{-\int P(s)ds} \, .$$

$\square$

**Proof.** Now we show some key steps to derive the solution,

$$y' + P(t)y = 0$$
$$\Rightarrow \quad \frac{dy}{dt} + P(t)y = 0 \quad \textit{complementary equation}$$
$$\Rightarrow \quad \frac{dy}{y} + P(t)dt = 0$$
$$\Rightarrow \quad \int \frac{dy}{y} + \int P(t)dt = 0$$
$$\Rightarrow \quad ln(y) = -\int P(t)dt + k$$
$$\Rightarrow \quad |y| = e^k e^{-\int P(s)ds}$$
$$\Rightarrow \quad c = e^k, \text{ if } y > 0 \text{ on } (a, b)$$
$$\Rightarrow \quad c = -e^k, \text{ if } y < 0 \text{ on } (a, b) \, .$$

$\square$

Denote the integrating factor $A(t) = y_1(t) = \exp\left(-\int P(s)ds\right)$, then the solution to Eq. (S.6) is

$$y = y_1(t) \left( \int \frac{Q(s)}{y_1(s)} ds + c \right) = A(t) \left( \int \frac{Q(s)}{A(s)} ds + c \right) \, .$$

**Lemma S.2.** *(Solution to the Initial Value Problem ([MacCluer et al., 2019](#)).) If $t_0$ is an arbitrary point in $(a, b)$ and $y_0$ is an arbitrary real number, then the initial value problem*

$$y' + P(t)y = Q(t), \quad y(t_0) = y_0. \tag{S.7}$$

*has the unique solution*

$$y = y_1(t) \left( \frac{y_0}{y_1(t_0)} + \int_{t_0}^{t} \frac{Q(s)}{y_1(s)} ds \right) .$$

□

## D.2   CLOSED-FORM SOLUTION OF THE LIF DYNAMICS AS A FIRST-ORDER LINEAR DIFFERENTIAL EQUATION

For the Analytical Closed-form Solution for the IVP of the LIF Dynamics, we have the following lemma in the main paper.

**Lemma 3.** *The analytical closed-form solution for the first-order IVP (Initial Value Problem) of the LIF dynamics ODE is as follows ([Gerstner et al., 2014](#)),*

$$U(t) = \exp\left(-\frac{t}{\tau}\right) \left( \int_0^t \frac{R}{\tau} I(s) \exp\left(\frac{s}{\tau}\right) ds + U_0 \right) . \tag{S.8}$$

**Proof.** If the injected current input $I(t)$ is a function depending on $t$, the LIF neuron dynamics becomes

$$\tau \frac{dU(t)}{dt} = -U(t) + RI(t) .$$

Write it into standard first-order differential equation

$$y' + P(t)y = Q(t) .$$

Consider the LIF neuron dynamics, and follow Lemma S.1

$$\frac{dU(t)}{dt} + \frac{1}{\tau}U(t) = \frac{R}{\tau}I(t)$$

$$\Rightarrow \quad P(t) = \frac{1}{\tau}, \quad Q(t) = \frac{R}{\tau}I(t)$$

$$\Rightarrow \quad A(t) = \exp\left(-\int P(t)dt\right) = \exp\left(-\frac{t}{\tau}\right) \quad (A(0) = 1)$$

$$\Rightarrow \quad U(t) = A(t)\left(\int \frac{Q(s)}{A(s)}ds + c\right)$$

$$\Rightarrow \quad U(t) = A(t)\left(\int \frac{R}{\tau}I(s)\frac{1}{A(s)}ds + c\right)$$

$$\Rightarrow \quad U(t) = \exp\left(-\frac{t}{\tau}\right)\left(\int \frac{R}{\tau}I(s)\exp\left(\frac{s}{\tau}\right)ds + c\right) .$$

If the initial condition $U(0) = U_0$, then $c = \frac{U_0}{A(0)} = U_0$, and by following Lemma S.2, the final closed-form solution is

$$U(t) = \exp\left(-\frac{t}{\tau}\right) \left( \int_0^t \frac{R}{\tau} I(s) \exp\left(\frac{s}{\tau}\right) ds + U_0 \right) .$$

□

*Note:* If the neuron starts at some value $U_0$ with no further current input, i.e., $I(t) = 0$, the solution of the linear differential equation above is as follows

$$U(t) = U_0 \exp\left(-\frac{t}{\tau}\right) .$$

In the absence of input current, the membrane potential will start at $U_0$ and exponentially decay with a time constant $\tau$. Therefore, we can determine the membrane potential ratio, often referred to as the leak factor, denoted by $\lambda$, as $\lambda = \frac{U(t+\Delta t)}{U(t)} = \frac{U_0 \exp\left(-\frac{t+\Delta t}{\tau}\right)}{U_0 \exp\left(-\frac{t}{\tau}\right)} = \exp\left(-\frac{\Delta t}{\tau}\right)$.

This relationship enables us to formulate the discretization scheme as $U[t+1] = \lambda U[t]$. This provides insights into the behavior of the membrane potential in the absence of input and establishes the discretization principle used for LIF modeling.

Let's consider a more practical case for the input current $I(t)$, where $I(t)$ is a function of $t$, not zero nor other constant values.

**Lemma 4.** *Through applying integration by parts, we derive another equivalent form of the closed-form solution for the LIF dynamics, denoted as:*

$$U(t) = \overbrace{(U_0 - RI_0)\exp\left(-\frac{t}{\tau}\right)}^{\text{exponential decay term}} + \overbrace{RI(t)}^{\text{input current model}} \underbrace{- \int_0^t R\exp\left(\frac{s-t}{\tau}\right) dI(s)}_{\text{absent in the discrete LIF model}} . \tag{S.9}$$

$$\underbrace{(U_0 - RI_0)\exp\left(-\frac{t}{\tau}\right) + RI(t)}_{\text{commonly considered in the discrete LIF model}}$$

*With the application of the Riemann–Stieltjes integral, the discretization version of the closed-form solution is represented as:*

$$U[t] = \overbrace{\lambda U[t-1]}^{(U_0 - RI_0)\exp\left(-\frac{t}{\tau}\right)} + \overbrace{X[t]}^{WO[t]=RI[t]} \underbrace{- \sum_{i=0}^{n} g_i X[s_i]}_{\text{TAB method}} . \tag{S.10}$$

**Proof.** For the term $\int_0^t \frac{R}{\tau} I(s)\exp\left(\frac{s}{\tau}\right) ds$, we can make some simplifications using rigorous mathematical derivation, and we have

$$\int_0^t \frac{R}{\tau} I(s)\exp\left(\frac{s}{\tau}\right) ds$$

$$= \int_0^t RI(s) d\exp\left(\frac{s}{\tau}\right)$$

$$= RI(s)\exp\left(\frac{s}{\tau}\right)\Big|_0^t - \int_0^t R\exp\left(\frac{s}{\tau}\right) dI(s)$$

$$= RI(t)\exp\left(\frac{t}{\tau}\right) - RI(0) - \int_0^t R\exp\left(\frac{s}{\tau}\right) dI(s) .$$

Then, we have

$$U(t) = \exp\left(-\frac{t}{\tau}\right)\left(\int_0^t \frac{R}{\tau} I(s)\exp\left(\frac{s}{\tau}\right) ds + U_0\right)$$

$$\Rightarrow \quad U(t) = \exp\left(-\frac{t}{\tau}\right)\left(RI(t)\exp\left(\frac{t}{\tau}\right) - RI(0) - \int_0^t R\exp\left(\frac{s}{\tau}\right) dI(s) + U_0\right)$$

$$\Rightarrow \quad U(t) = RI(t) + (U_0 - RI(0))\exp\left(-\frac{t}{\tau}\right) - \exp\left(-\frac{t}{\tau}\right)\int_0^t R\exp\left(\frac{s}{\tau}\right) dI(s)$$

$$\Longleftrightarrow \quad U(t) = (U_0 - RI(0))\exp\left(-\frac{t}{\tau}\right) + RI(t) - \int_0^t R\exp\left(\frac{s-t}{\tau}\right) dI(s)$$

$$\Rightarrow \quad U(t) = \underbrace{(U_0 - RI(0))\exp\left(-\frac{t}{\tau}\right) + RI(t)}_{\text{considered in the commonly used discrete LIF model}} \underbrace{- \int_0^t R\exp\left(\frac{s-t}{\tau}\right) dI(s)}_{\text{not in the discrete LIF model}} .$$

Finally, we have derived the closed-form solution of the LIF model,

$$U(t) = \underbrace{(U_0 - RI(0))\exp\left(-\frac{t}{\tau}\right)}_{\text{exponential decay term}} + \underbrace{RI(t)}_{\text{injected input currents}} \underbrace{- \int_0^t R\exp\left(\frac{s-t}{\tau}\right) dI(s)}_{\text{not in the discrete LIF model}} \tag{S.11}$$

Specifically,

$$U(t) = \overbrace{(U_0 - RI_0)\exp\left(-\frac{t}{\tau}\right)}^{\text{exponential decay term}} + \overbrace{RI(t)}^{\text{input current model}} \underbrace{- \int_0^t R\exp\left(\frac{s-t}{\tau}\right) dI(s)}_{\text{absent in the discrete LIF model}} . \qquad \text{(S.12)}$$

$$\underbrace{\hphantom{(U_0 - RI_0)\exp\left(-\frac{t}{\tau}\right) + RI(t)}}_{\text{commonly considered in the discrete LIF model}}$$

□

As the third term is a bit complicated with an integral over time, we cannot drop it without any declaration for just simplification. To understand more about the third term, we consider applying integration by parts, and derive a discrete version of the closed-form solution as it can be used in a spiking neural network.

### D.3 DISCRETIZING THE CONTINUOUS CLOSED-FORM SOLUTION OF THE LIF MODEL

By the definition of the Riemann–Stieltjes integral, we can discretize the continuous integral by replacing the integral with Riemann sum.

Denote a partition of an interval $[0, t]$ with a finite sequence of numbers of the form $0 = s_0 < s_1 < \cdots < s_i < \cdots < s_n = t$.

The Riemann sum of the function with respect to the tagged partition $s_0, \cdots, s_n$ is

$$\int_0^t R\exp\left(\frac{s-t}{\tau}\right) dI(s)$$

$$\approx \sum_{i=0}^{n-1} \exp\left(\frac{s_i - t}{\tau}\right) * (RI[s_{i+1}] - RI[s_i]) \quad (\text{ denote } X[s_i] = RI[s_i])$$

$$= \sum_{i=0}^{n-1} \exp\left(\frac{s_i - t}{\tau}\right) * (X[s_{i+1}] - X[s_i])$$

$$= \exp\left(\frac{s_0 - t}{\tau}\right)(X[s_1] - X[s_0]) + \exp\left(\frac{s_1 - t}{\tau}\right)(X[s_2] - X[s_1])$$

$$\quad + \cdots + \exp\left(\frac{s_{n-1} - t}{\tau}\right)(X[s_n] - X[s_{n-1}])$$

$$= \exp\left(\frac{s_0 - t}{\tau}\right)X[s_0] + \left\{\exp\left(\frac{s_0 - t}{\tau}\right) - \exp\left(\frac{s_1 - t}{\tau}\right)\right\}X[s_1] + \exp\left(\frac{s_n - t}{\tau}\right)X[s_n]$$

$$= \sum_{i=0}^{n} g_i X[s_i] \quad \text{where } g_i = \exp\left(\frac{s_{i-1} - t}{\tau}\right) - \exp\left(\frac{s_i - t}{\tau}\right) \text{ for } i = 1, \cdots, n-1, \text{ and } g_i < 0.$$

$$g_0 = \exp\left(\frac{s_0 - t}{\tau}\right) > 0, g_n = \exp\left(\frac{s_n - t}{\tau}\right) = 1 .$$

Note that $g_0 > 0$ and $g_n = 1$, weights $g_i$ can be positive or negative for $i = 1, \cdots, n-1$. It makes no difference for positive or negative weights, as the signs can be absorbed into the learnable parameters of the TAB method.

Finally, from the continuous version of the closed-form solution of the LIF model,

$$U(t) = (U_0 - RI(0))\exp\left(-\frac{t}{\tau}\right) + RI(t) - \int_0^t R\exp\left(\frac{s-t}{\tau}\right) dI(s) ,$$

we have the discrete version of the closed-form solution of the LIF model,

$$U[t] = (U_0 - RI[0])\exp\left(-\frac{t}{\tau}\right) + RI[t] - \sum_{i=0}^{n} g_i X[s_i] .$$

In summary, we have

$$U(t) = (U_0 - RI(0))\exp\left(-\frac{t}{\tau}\right) + RI(t) - \int_0^t R\exp\left(\frac{s-t}{\tau}\right) dI(s) \tag{S.13}$$

$$U[t] = \lambda U[t-1] + RI[t] - \sum_{i=0}^{n} g_i X[s_i] \tag{S.14}$$

$$U[t] = \lambda U[t-1] + X[t] - \sum_{i=0}^{n} g_i X[s_i] . \tag{S.15}$$

Then we have the discrete version of the closed-form solution of the LIF model,

$$U[t] = \overbrace{\lambda U[t-1]}^{(U_0 - RI_0)\exp\left(-\frac{t}{\tau}\right)} + \overbrace{X[t]}^{WO[t]=RI[t]} \underbrace{- \sum_{i=0}^{n} g_i X[s_i]}_{\textbf{\textcolor{blue}{TAB method}}} . \tag{S.16}$$

In this formulation, the first exponential decay term, $\lambda U[t-1]$, captures the temporal dependency of the membrane potential from the preceding time-step. The second term, a simple current input model, $RI[t] = WO[t]$, incorporates spikes from the pre-connected neurons at the current time-step $[t]$. Significantly, the third term, representing the temporal accumulated input across all previous time-steps through a weighted sum of the input currents $X[s_i]$ with associated weights $g_i$, introduces a novel concept. This accumulation mechanism of the inputs is a foundational component of the TAB method, providing a link that connects the TAB method and the neural dynamics.

Comparing the commonly used discrete LIF model with the discrete closed-form solution, we observe the absence of the accumulation term in the former. Our TAB method reintroduces this accumulation term in the normalization procedure. In this paper, we still employ the commonly used discrete LIF neuron model where the third term $-\sum_{i=0}^{n} g_i X[s_i]$ is simplified/omitted, but in the TAB method, we consider this term back during the normalization step.

The TAB method normalizes the data using *temporal accumulated* information from time-step 1 to $t$, which takes into account historical temporal dependencies but does not look ahead to future temporal information. This adjustment also brings the discrete LIF model closer to the analytical closed-form counterpart and establishes a natural connection between neuron dynamics and batch normalization.

# E  EXPERIMENTS DETAILS

## E.1  OPERATING ENVIRONMENTS

All of our models are trained on the PyTorch platform. Experiments are conducted on an NVIDIA RTX A6000 GPU.

## E.2  DATASETS

The CIFAR dataset (Krizhevsky et al., 2009) consists of $50,000$ training images and $10,000$ testing images each with the size of $32 \times 32$. There are 10 classes in CIFAR-10 dataset and 100 classes in CIFAR-10 dataset.

**CIFAR-10**: The CIFAR-10 dataset (Krizhevsky et al., 2009) consists of $60,000$ color images each with image size of $32 \times 32$ in 10 classes of objects such as airplanes, cars, and birds, with $6,000$ images per class. There are $50,000$ samples in the training set and $10,000$ samples in the test set.

**CIFAR-100**: The CIFAR-100 dataset (Krizhevsky et al., 2009) consists of $60,000$ $32 \times 32$ color images in 100 classes with $6,000$ images per class. There are $50,000$ samples in the training set and $10,000$ samples in the test set.

**DVS-CIFAR10**: DVS-CIFAR10 (Li et al., 2017), a challenging mainstream neuromorphic data set, is used in this study. DVS-CIFAR10 is converted from CIFAR10. The DVS-CIFAR10 consists of $10,000$ images with size $128 \times 128$.

### E.3 DATA PRE-PROCESSING

For CIFAR-10 and CIFAR-100 datasets, common data normalization and some data pre-processing techniques are used in the experiments. For example, we resize the images in the CIFAR-10/CIFAR-100 datasets into $32 \times 32$. We apply data normalization to the CIFAR-10 and CIFAR-100 datasets. We apply random horizontal flipping and random cropping to the training images as data augmentation. We use these above data pre-processing techniques on the datasets in comparison experiments with other BN methods in SNNs. In addition, we add cutout operations (DeVries & Taylor, 2017) as TEBN (Duan et al., 2022) and Dspike (Li et al., 2021b) did on ResNet, when comparing our TAB method with other state-of-the-art methods.

Our DVS-CIFAR10 dataset is loaded from the Spikingjelly framework (Fang et al., 2020). Following Samadzadeh et al. (2023); Duan et al. (2022), we split the dataset into $9,000$ training images and $1,000$ testing images, and reduce the spatial resolution from the original $128 \times 128$ to $48 \times 48$.

### E.4 NETWORK ARCHITECTURES AND TRAINING CONFIGURATIONS

In comparison experiments with existing BN methods, we reproduce the same architectures as BNTT (Kim & Panda, 2021) and TEBN (Duan et al., 2022): VGG-9 network architecture (64C3-64C3-AP2-128C3-128C3-AP2-256C3-256C3-256C3-AP2-1024FC-10FC) on CIFAR-10, VGG-11 (64C3-128C3-AP2-256C3-256C3-AP2-512C3-512C3-AP2-512C3-512C3-AP2-4096FC-100FC) on CIFAR-100, and 7-layer CNN (64C3-AP2-128C3-128C3-AP2-256C3-256C3-AP2-1024FC-10FC) on DVS-CIFAR10. Following TEBN (Duan et al., 2022), we add Dropout (Srivastava et al., 2014) layers before fully-connected layers to enhance generalization, but we do not use a voting layer after the last fully-connected layer in DVS-CIFAR10 experiments as in Duan et al. (2022). We use the standard ResNet-19 for both CIFAR-10 and CIFAR-100.

The Stochastic Gradient Descent (SGD) optimizer (Bottou, 2012) is used in the experiments with a momentum parameter of $0.9$. We use a cosine decay scheduler (Loshchilov & Hutter, 2017) to adjust the learning rate with a weight decay $5 \times 10^{-4}$ for CIFAR-10/CIFAR-100 datasets. All models are trained for 200 epochs. We set the initial learning rate to $\epsilon = 0.01$ for CIFAR-10 and CIFAR-100.

As for the input to the first layer and the output of the last layer of the SNN, we do not employ any spiking mechanism as in Li et al. (2021a); Bu et al. (2022). We use constant input when evaluating the SNNs. We directly encode the static image to temporal dynamic spikes as input to the first layer, which can prevent the undesired information loss introduced by the Poisson encoding. For the last layer output, we only integrate the pre-synaptic inputs and do not fire any spikes, so that the output neurons only output the temporal average of the pre-synaptic inputs of all time-steps $(m_1, m_2, \cdots, m_C)$ with $C$ as the number of classes. The output is then fed into a Softmax layer to compute the cross-entropy loss with the true labels $(y_1, y_2, \cdots, y_C)$,

$$L_{CE} = -\sum_{i=1}^{C} y_i \log \left( \frac{e^{m_i}}{\sum_{j=1}^{C} e^{m_j}} \right) .$$

To allow for effective BP training, the triangle-shaped surrogate gradients are used

$$\frac{\partial \boldsymbol{o}_i[t]}{\partial \boldsymbol{u}_i[t]} = \max \left\{ 0, 1 - |\boldsymbol{u}_i[t] - \theta| \right\} .$$

Other hyperparameters can be found in Table S2.

Table S2: Hyperparameters optimization for training

| Dataset | Optimizer | Scheduler | Epochs | Learning Rate | Batch Size |
|---|---|---|---|---|---|
| CIFAR-10 | SGD | CosineAnnealingLR(T=200) | 200 | 0.02 | 64 |
| CIFAR-100 | SGD | CosineAnnealingLR(T=200) | 200 | 0.02 | 64 |
| DVS-CIFAR10 | SGD | CosineAnnealingLR(T=200) | 200 | 0.05 | 64 |

### E.5 MORE RESULTS

We compare our TAB method with other state-of-the-art approaches in SNNs, including RMP (Han et al., 2020), ReLU-Threshold-Shift (RTS) (Deng & Gu, 2021), quantization clip-floor-shift (QCFS) (Bu et al., 2022), progressive tandem learning (PTL) (Wu et al., 2021b), Hybrid Conversion (HC) (Rathi et al., 2020), Temporal-Coded method (TC) (Zhou et al., 2021), Temporal Spike Sequence Learning Back-Propagation (TSSL-BP) (Zhang & Li, 2020), Differentiable Spike (Dspike) (Li et al., 2021b), Temporal Efficient Training (TET) (Deng et al., 2022), TEBN (Duan et al., 2022). We compare our method with other state-of-the-art learning methods for SNNs on CIFAR-100, and DVS-CIFAR10 datasets and report the results in Table S3. Results on ImageNet dataset are reported in Table S4, with other state-of-the-art learning methods.

Table S3: Comparison between the proposed TAB and other state-of-the-art approaches in SNNs.

| Dataset | Model | Method | Architecture | Time-steps | Accuracy (%) |
|---|---|---|---|---|---|
| CIFAR-100 | RMP (Han et al., 2020) | ANN-to-SNN | ResNet-20 | 2048 | 67.82 |
| | RTS (Deng & Gu, 2021) | ANN-to-SNN | ResNet-20 | 512 | 72.34 |
| | QCFS (Bu et al., 2022) | ANN-to-SNN | ResNet-20 | 128 | 70.55 |
| | HC (Rathi et al., 2020) | Hybrid Training | VGG-11 | 2500 | 70.94 |
| | Dspike (Li et al., 2021b) | Surrogate Gradient | ResNet-18 | 6 / 4 / 2 | 74.24 / 73.35 / 71.68 |
| | TET (Deng et al., 2022) | Surrogate Gradient | ResNet-19 | 6 / 4 / 2 | 74.72 / 74.47 / 72.87 |
| | TEBN (Duan et al., 2022) | Surrogate Gradient | VGG-11 | 4 | 74.37 |
| | TEBN (Duan et al., 2022) | Surrogate Gradient | ResNet-19 | 6 / 4 / 2 | 76.41 / 76.13 / 75.86 |
| | **TAB (Ours)** | Surrogate Gradient | VGG-11 | 4 | **75.89** |
| | | | ResNet-19 | 6 / 4 / 2 | **76.82 / 76.81 / 76.31** |
| DVS-CIFAR10 | PLIF (Fang et al., 2021b) | Surrogate Gradient | 6-layer CNN | 20 | 74.80 |
| | Dspike (Li et al., 2021b) | Surrogate Gradient | ResNet-18 | 10 | 75.40 |
| | TET (Deng et al., 2022) | Surrogate Gradient | VGGSNN | 10 | 83.17 |
| | TEBN (Duan et al., 2022) | Surrogate Gradient | 6-layer CNN | 10 | 80.00 |
| | TEBN (Duan et al., 2022) | Surrogate Gradient | VGGSNN | 10 | 84.90 |
| | **TAB (Ours)** | Surrogate Gradient | 7-layer CNN | 4 | **76.7** |
| | | | 6-layer CNN | 2 | **84.21** |

Table S4: Comparison between the proposed TAB and other state-of-the-art approaches on ImageNet.

| Model | Method | Architecture | Time-steps | Accuracy (%) |
|---|---|---|---|---|
| SPIKE-NORM (Sengupta et al., 2019) | ANN-to-SNN | ResNet-34 | 2500 | 69.96 |
| RTS (Deng & Gu, 2021) | ANN-to-SNN | VGG-16 | 16 | 55.80 |
| QCFS (Bu et al., 2022) | ANN-to-SNN | ResNet-34 | 16 | 59.35 |
| SlipReLU (Jiang et al., 2023) | ANN-to-SNN | ResNet-34 | 32 | 66.61 |
| SNNC-AP (Li et al., 2021a) | ANN-to-SNN | ResNet-34 | 32 | 64.54 |
| Hybrid Conversion (Rathi et al., 2020) | Hybrid Training | ResNet-34 | 250 | 61.48 |
| TET (Deng et al., 2022) | Surrogate Gradient | Spiking-ResNet-34 | 6 | 64.79 |
| tdBN (Duan et al., 2022) | Surrogate Gradient | ResNet-34 | 6 | 63.72 |
| TEBN (Duan et al., 2022) | Surrogate Gradient | ResNet-34 | 4 | 64.29 |
| **TAB (Ours)** | Surrogate Gradient | ResNet-34 | 4 | **67.78** |
| | | ResNet-34 | 2 | **65.94** |

## F THE OPTIMIZING OBJECTIVE FUNCTION OF BN METHODS IN SNNS

In the main paper, we have developed the TAB method to address the temporal covariate shift problem by investigating the neuron dynamics, specifically by aligning with the accumulated membrane potential. The TAB method utilizes the temporal accumulated statistics for normalization. As the accumulation process aligns naturally with the membrane potential accumulation procedure, the TAB method is conceived.

In fact, we can take a more general approach by directly exploring the additional temporal dimension, which opens up possibilities for developing different variants of BN methods for SNNs. To facilitate this adaptability and the development of a range of BN variants in SNNs, we introduce an optimizing objective function that highlights the distinctions in defining BN methods along the time dimension. This framework empowers the creation of a broader spectrum of BN methods tailored to the unique characteristics of SNNs.

We draw inspiration from BN techniques used in ANNs (Ioffe & Szegedy, 2015; Lian & Liu, 2019), and we establish a rigorous mathematical optimization objective function for BN methods in SNNs. Given the presence of an additional temporal/time dimension in SNNs,it is imperative to handle the time dimension meticulously when formulating the objective function for BN in SNNs. Refer to Sect. 5 for notation details.

Let us define the normalization operator $f_W^{B,\phi}(\cdot)$ as the mapping of a function to a function, associated with a mini-batch data samples $B$, an activation function $\phi$, and parameters $W$. Additionally, we introduce the function $g(\cdot)$ defined as $g(\cdot) = (g_1(\cdot), \cdots, g_n(\cdot), 1)^T$, where each $g_i(\cdot)$ maps a tensor to a scalar value. The normalization operator $f_W^{B,\phi}(\cdot)$ can be defined by

$$f_W^{B,\phi} \circ g(\cdot) := f_W^{B,\phi}(g)(\cdot) = \phi \left( W \left( \frac{g_1 - \boldsymbol{mean}(g_1, B)}{\boldsymbol{var}(g_1, B)}, \cdots, \frac{g_n - \boldsymbol{mean}(g_n, B)}{\boldsymbol{var}(g_n, B)}, 1 \right)^T \right)$$

where $\boldsymbol{mean}(g_i, B) := \frac{1}{|B|} \sum_{b \in B} g_i(b)$ and $\boldsymbol{var}(g_i, B) := \boldsymbol{mean}(g_i^2, B) - (\boldsymbol{mean}(g_i, B))^2$. Note that the argument of $f_W^{B,\phi}$ is a function $g(\cdot)$, and $f_W^{B,\phi} \circ g$ is another function, the first argument of $\boldsymbol{mean}(g_i, B)$ and $\boldsymbol{var}(g_i, B)$ is a function and the second augment is a set of sample from the mini-batch $B$.

An $m$-layer spiking neural network with BN can be represented by a function

$$F_{\boldsymbol{W}}^B(\cdot) := F_{\{W_j\}_{j=1}^m}^B(\cdot) := f_{W_m}^{B,\phi_m} \circ f_{W_{m-1}}^{B,\phi_{m-1}} \circ \cdots \circ f_{W_1}^{B,\phi_1} \circ I(\cdot) \, ,$$

where $I(\cdot)$ is the identical mapping function $I(\boldsymbol{x}) = \boldsymbol{x}$, and $\boldsymbol{W} := \{W_j\}_{j=1}^m$ denote all the learnable weights in the neural network, and $\phi_j$'s are the activation functions for each layer. Using similar notations, we can represent an SNN without BN by another function $\tilde{f}_W^\phi$

$$F_{\boldsymbol{W}}(\cdot) := F_{\{W_j\}_{j=1}^m}(\cdot) := \tilde{f}_{W_m}^{\phi_m} \circ \tilde{f}_{W_{m-1}}^{\phi_{m-1}} \circ \cdots \circ \tilde{f}_{W_1}^{\phi_1} \circ I(\cdot) \, ,$$

where $\tilde{f}_W^\phi$ is defined by $\tilde{f}_W^\phi \circ g(\cdot) = \phi(Wg(\cdot))$.

Besides the difference of operator functions $\tilde{f}_{W_m}^{\phi_m}$ and $f_{W_m}^{B,\phi_m}$, their optimization objective functions with BN and without BN are also different.

Given the training dataset $\mathfrak{D}$, the optimization objective function without BN is defined by

$$\min_{\boldsymbol{W}} \frac{1}{|\mathfrak{D}|} \sum_{(\boldsymbol{x}[t],y) \in \mathfrak{D}_t} L_{CE} \left( \frac{1}{T} \sum_{t=1}^T F_{\boldsymbol{W}}(\boldsymbol{x}[t]), y \right) \, , \tag{S.17}$$

where $L_{CE}(\cdot, \cdot)$ is a predefined Cross-Entropy loss function, $\mathfrak{D}_t$ is spiking representation at time-step $t$, $\mathfrak{D}$ is the training dataset consisting of spiking representations from all time-steps. Specially

$$\mathfrak{D}_t = \{(\boldsymbol{x}[t], y), \text{ with } \boldsymbol{x}[t] = SpikeRep(\boldsymbol{x}, t) \text{ and } \forall \, (\boldsymbol{x}, y) \in \mathcal{D}\} \, .$$

At each time-step $t$, we feed spiking data $\boldsymbol{x}[t]$ into the network $F_{\boldsymbol{W}}(\cdot)$, which produces output $F_{\boldsymbol{W}}(\boldsymbol{x}[t])$. The final prediction is obtained by aggregating results over the time dimension, resulting in $\hat{y} = \frac{1}{T} \sum_{t=1}^T F_{\boldsymbol{W}}(\boldsymbol{x}[t])$.

The optimization objective function with BN is

$$(\textbf{BN}) \quad \min_{\boldsymbol{W}} \frac{1}{|\mathfrak{B}|} \sum_{B \in \mathfrak{B}} \frac{1}{|B|} \sum_{(\boldsymbol{x}[t],y) \in \mathcal{B}_t} L_{CE} \left( \frac{1}{T} \sum_{t=1}^T F_{\boldsymbol{W}}^B(\boldsymbol{x}[t]), y \right) \, , \tag{S.18}$$

where $\mathcal{B}_t \subset \mathfrak{D}_t$ represents a subset of the dataset at time-step $t$, and $\mathfrak{B}$ is a partition of the whole dataset $\mathfrak{D}$, that is, the set of mini-batch sets, with $\sum_{B \in \mathfrak{B}} |B| = |\mathfrak{D}|$. The mini-batch set $B$ is used to

calculate the mini-batch mean and variance. The objective function of SNN with BN is very different from that without BN, due to the inclusion of BN.

The presence of the additional temporal dimension in SNNs gives rise to different BN variants, depending on the chosen approaches for defining mini-batches along the time dimension. Therefore, BN methods in SNNs are more varied than in ANNs because of the additional temporal dimension in SNNs.

The mini-batch sets $B$ in SNNs differ significantly from those in ANNs due to the presence of the additional temporal dimension, as the mini-batch set $B$ is used to calculate the mini-batch mean and variance. Specifically, the mini-batch sets $B$ can be viewed as a partition function of the subset $\{\mathcal{B}_t\}_{t=1}^T$ along the temporal dimension, represented as

$$B = r(\mathcal{B}_1, \mathcal{B}_2, \cdots, \mathcal{B}_t, \cdots, \mathcal{B}_T) .$$

Denote the partition $\mathfrak{B} = \{B \subset \mathfrak{B}, B = r(\mathcal{B}_1, \cdots, \mathcal{B}_t, \cdots, \mathcal{B}_T), \mathcal{B}_t \subset \mathfrak{D}_t\}$. Moreover, it is noteworthy that at each time-step $t$, the data $(\boldsymbol{x}[t], y)$ always come from $\mathcal{B}_t$.

The definition and partitioning of mini-batches along the time dimension are critical factors that shape BN methods in SNNs and their associated objective functions. This leads us to the following important observation: ***Different partition/sampling strategies along the temporal dimension give rise to distinct BN methods in SNNs***.

### F.1 SPECIAL CASES OF THE OPTIMIZING OBJECTIVE FUNCTION

For the optimizing objective function, the mini-batch set $B$ (or the sampling strategy) plays an important role, as it is used to calculate the mini-batch mean and variance. Research works on BN methods in SNNs primarily differ in their partition strategies along the temporal dimension. Notably, the proposed optimizing objective function encompasses most existing BN methods in SNNs as special cases, including our proposed TAB method. Refer to Append. F.2 for details.

In the BNTT method (Kim & Panda, 2021), data are normalized ***independently*** at each time-step $t$, where the mini-batch $B$ is defined ***temporal-independently*** with data at each time-step $t$, i.e., $B \triangleq \mathcal{B}_t$. In TEBN (Duan et al., 2022) and tdBN (Zheng et al., 2021), data are normalized ***temporal-jointly*** across ***all $T$ time-steps***. The mini-batch $B$ is defined with data across all $T$ time-steps, i.e., $B \triangleq B_{1:T}$. However, the underlying assumption of sampling strategy $B_{1:T}$ *implicitly assumes* that at each time-step $t$, we have access to data spanning all $T$ time-steps, including the current time-step $t < T$ even future time-steps $T$. Nevertheless, this assumption may not hold true in practical situations. In contrast, our proposed TAB method takes a different partition approach. It employs a strategy to ***jointly*** normalize the data using ***temporal accumulated*** information from time-step 1 to $t$, denoted as $B \triangleq B_{1:t}$. By incorporating the temporal accumulated information, TAB ensures efficient utilization of all available information up to the current time-step $t$, while it does not look ahead to the future information beyond $t$ such as $T$. The data normalization process in TAB follows an ***accumulative-joint*** scheme, which aligns with the analytical closed-form solution of the neuron dynamics.

### F.2 SPECIAL CASES OF BN METHODS IN SNNS

For the optimizing objective function of BN methods for SNNs, the mini-batch set $B$ (or the sampling strategy) plays an important role, as it is used to calculate the mini-batch mean and variance. The research works on BN methods in SNNs primarily differ in their approaches to sampling strategies. Notably, the proposed optimizing objective function encompasses most existing BN methods in SNNs as special cases, including our proposed TAB method.

Current research works on batch normalization methods for SNNs differ from each other from the perspective of sampling strategies.

- A typical $\mathfrak{B}$ can be defined **temporal-independently** using data at each time-step $t$, i.e.,

  $B = \mathcal{B}_t$, by $\mathfrak{B} = \bigcup B := \bigcup_{t=1}^T \{\mathcal{B}_t\} = \{\mathcal{B}_1, \mathcal{B}_2, \cdots, \mathcal{B}_T\}$ with $\mathcal{B}_t \subset \mathfrak{D}_t$, where

  $$B = \mathcal{B}_t = \{(\boldsymbol{x}[t], y), \text{ where } \boldsymbol{x}_t \in \mathbb{R}^{C \times H \times W}, \forall (\boldsymbol{x}[t], y) \in \mathfrak{D}_t\} .$$

As $\mathfrak{D}_t \cap \mathfrak{D}_s = \emptyset$ if $t \neq s$, then $\mathcal{B}_t \cap \mathcal{B}_s = \emptyset$ if $t \neq s$, and $\mid \mathfrak{B} \mid = T$. As the mini-batch set $B$ only uses $\mathcal{B}_t$, we denote $B \triangleq B_t$. The objective function becomes,

$$(\textbf{BN}) \quad \min_{\boldsymbol{W}} \frac{1}{T} \sum_{B_t \in \mathfrak{B}} \frac{1}{\mid B_t \mid} \sum_{(\boldsymbol{x}[t], y) \in \mathcal{B}_t} L_{CE}\left(\frac{1}{T} \sum_{t=1}^{T} F_{\boldsymbol{W}}^{B_t}(\boldsymbol{x}[t]), \, y\right) ,$$

By using this sampling strategy, Kim & Panda (2021) proposed the BNTT method, in which the data are normalized *independently* at each time-step $t$.

- Another common construction of $\mathfrak{B}$ is defined **temporal-jointly** using information from all $T$ time-steps, by $\mathfrak{B} = \bigcup B$ with $B \subset \mathfrak{D}$, where

$$B = \{(\boldsymbol{x}_{1:T}, y), \text{ where } \boldsymbol{x}_{1:T} = (\boldsymbol{x}[1]^\top, \boldsymbol{x}[2]^\top, \cdots, \boldsymbol{x}[T]^\top) \in \mathbb{R}^{T \times C \times H \times W}, \forall (\boldsymbol{x}[t], y) \in \mathfrak{D}_t\}.$$

As the mini-batch set $B$ uses all $\mathcal{B}_t$ at all time-steps from $t = 1$ to $t = T$, we denote $B \triangleq B_{1:T}$. The objective function becomes,

$$\min_{\boldsymbol{W}} \sum_{B_{1:T} \in \mathfrak{B}} \frac{1}{\mid B_{1:T} \mid} \sum_{(\boldsymbol{x}[t], y) \in \mathcal{B}_t} L_{CE}\left(\frac{1}{T} \sum_{t=1}^{T} F_{\boldsymbol{W}}^{B_{1:T}}(\boldsymbol{x}[t]), \, y\right) ,$$

Using this sampling strategy, researchers have proposed the TEBN method (Duan et al., 2022) and the tdBN method (Zheng et al., 2021). In these methods, the data are *temporal-jointly* normalized across all time-steps $T$.

However, choosing $\mathfrak{B}$ in this way *implicitly assumes* that all nodes at each time-step $t$ can access the same dataset over all $T$ time-steps, which may not be true in practice.

Another possible $\mathfrak{B}$ can be defined **jointly** using the **temporal accumulated** information from all previous time-steps up to the current time-step $t$, represented as $\mathfrak{B} := \{B_1, B_{1:2}, \cdots, B_{1:t}\}$ with

$$B = B_{1:t} := \{B_1 \cup \cdots \cup B_t, B_t \subset \mathcal{D}_t\}.$$

The objective function becomes,

$$\min_{\boldsymbol{W}} \frac{1}{t} \sum_{B_{1:t} \in \mathfrak{B}} \frac{1}{\mid B_{1:t} \mid} \sum_{(\boldsymbol{x}[t], y) \in \mathcal{B}_t} L_{CE}\left(\frac{1}{T} \sum_{t=1}^{T} F_{\boldsymbol{W}}^{B_{1:t}}(\boldsymbol{x}[t]), \, y\right) ,$$

From an optimization perspective, this sampling strategy aligns with our proposed TAB method, where TAB normalizes the data using the accumulated information over all time-steps from $1$ to $t$, i.e., $B := B_{1:t}$.

Moreover, the optimizing objective function highlights the distinctions in defining BN methods in SNNs along the time/temporal dimension, by which we can optimize and develop more BN variants along the time dimension.

## G  LIMITATIONS AND SOCIETAL IMPACT

While achieving better performance compared to existing methods, our method has only been tested on rate-encoding-based SNNs with surrogate gradients for back-propagation. Rate-encoding is popularly used in conventional SNNs where each neuron fires at most once, which is very different from timing-based encoding. Since timing-based encoding methods encode information in the relative timing across individual neurons without single-spike restrictions. Therefore, they allow each neuron to fire multiple times and bring better asynchronism. In our future research, we intend to investigate the application of our method within timing-based SNNs

In terms of social impacts, our research primarily focuses on the direct training of high-performance and low-latency SNNs, which does not bring obvious negative effects. Furthermore, with the increasing adoption of SNNs due to their energy-saving consumption properties, SNNs will become indispensable in edge computing applications.

