# OpenReview forum: "TAB: Temporal Accumulated Batch Normalization in Spiking Neural Networks"
_ICLR.cc/2024/Conference — ICLR 2024 poster_

### Official Review · Reviewer_vi7C · 2023-10-25

**Soundness:** 2 fair
**Presentation:** 3 good
**Contribution:** 2 fair
**Rating:** 6
**Confidence:** 4

**Summary:**

The paper discusses the challenges of training SNNs, such as non-differentiability of the spiking activation function and the temporal nature of data. It proposes a method called Temporal Accumulated Batch Normalization (TAB) which uses temporal accumulated statistics for data normalization to solve a well-established problem called Temporal Covariate Shift, enhancing the efficiency of SNN training. The paper presents experimental results on CIFAR-10, CIFAR-100, and DVS-CIFAR10 datasets, demonstrating the effectiveness of TAB in improving accuracy. The paper's strength lies in its novelty, detailed explanation of the method, and well-presented experimental results.

**Strengths:**

This paper has the following strengths:

The authors relate the proposed algorithm with the closed-form of the LIF dynamics and give principled insights into what components in SNN should be normalized.

The authors obtain better performance than previous algorithms.

The proposed method only records the past information, which can be some basic support to online training.

**Weaknesses:**

The paper lacks a detailed analysis of the computational complexity of TAB, which could be crucial for practical applications. Although it mentions that TAB requires more computations than other normalization techniques, it does not provide a detailed cost analysis or the impact of different hyperparameters on computational complexity, making it difficult to evaluate its practicality in real-world applications.

There seems to be a gradual improvement on the SNN batch normalization technique with the suggested method.

**Questions:**

What is the impact of TAB on energy efficiency, and how does it compare to other methods for training SNNs? While the authors mention that SNNs are attractive for their energy-efficient computing when implemented on neuromorphic hardware, they do not provide a detailed analysis of the impact of TAB on energy efficiency. This is an important question, as it would help to assess the potential impact of TAB on energy efficiency and identify potential areas for future research.




Can you provide a more detailed analysis of the computational complexity of TAB and how it compares to other methods?




Can you provide evidence that TAB solves TCS? I think the illustration of distribution is capable of demonstrating the solution.

---

> ### Author Response · Authors · 2023-11-22
> **Response to Reviewer vi7C (part 1/2)**
>
> > 1. A detailed analysis of the impact of TAB on energy efficiency
>
> **Response 1:**
> Thank you for your valuable feedback. In response to your suggestion, we have examined the average firing rate, specifically focusing on activity sparsity, to illustrate the network's energy efficiency. Detailed plots illustrating this analysis can be found [here](https://anonymous.4open.science/r/ICLR2024_TAB_Rebuttal-BF2C/spike_density_plot.pdf).
>
> Upon careful examination of the plots, it is evident that the average spiking firing rate hovers around 10%-20%. This observation implies that merely 10%-20% of the neurons are actively firing spikes. In practical terms, this means that only a fraction of the neurons, approximately 10%-20%, are utilized for transmitting information between layers. This level of sparsity underscores the network's remarkable efficiency in transmitting information through the utilization of a minimal subset of neurons. We believe these findings significantly contribute to the energy-efficient nature of our proposed approach.
>
>
> > 2. Can you provide a more detailed analysis of the computational complexity of TAB and how it compares to other methods?
>
>
> **Response 2:**
> Thanks for the advice. The Table summarizes the Complexity Cost of different BN methods, including computational complexity and Memory Cost.
>
>
> | BN methods | Computational Complexity | Memory Cost |
> | -------|------------| -------------|
> | Conventional BN | $\mathcal{O}(BCHW) $  |  $\mathcal{O}(CHW)$ |
> | BNTT [1] | $\mathcal{O}(TBCHW)$ |  $\mathcal{O}(TCHW)$ |
> | tdBN [2] | $\mathcal{O}(TBCHW)$ | $\mathcal{O}(CHW)$ |
> | TEBN [3] | $\mathcal{O}(TBCHW)$  |  $\mathcal{O}(CHW) + \mathcal{O}(T)$ |
> | TAB (ours) | $\mathcal{O}(TBCHW)$ |  $\mathcal{O}(CHW) + \mathcal{O}(T)$ |
>
>
> - (1) We have obtained this Complexity analysis based on the statistics and parameters of different BN methods in Table 1 in the main paper. We use here T to refer to the number of time-steps, B to refer to the batch size, C,H,W are the number of channels, height, and width. So the input at time-step t, $X_t$, to the normalization layer in SNNs has a dimension of $\mathbf{R}^{C \times H \times W}$.
>
> - (2) In the conventional BN method in ANNs, there is no time dimension, therefore, the traditional BN method in ANNs has the computational complexity of $\mathcal{O}(BCHW) $, and a Memory Cost of $\mathcal{O}(CHW)$ to save the mean and variance which have the same dimension as $X$.
>
>
> **Memory Cost:**
> - a) As BNTT independently normalizes data at each time-step, it needs $\mathcal{O}(TCHW)$ space to save the mean and variance at each time-step, which is T times more than the traditional BN method, tdBN, and TEBN.
> - b) The tdBN and TEBN jointly normalize data across all time-steps, so they only need $\mathcal{O}(CHW)$ space to save one overall mean and variance which is the same as the traditional BN method for saving mean and variance.
> - c) The total memory cost of tdBN is $\mathcal{O}(CHW)$, as it needs only shared overall batch parameters just as the traditional BN, so there is no additional memory cost for saving the batch parameters.
> - d) The total memory cost of TEBN is $\mathcal{O}(CHW) + \mathcal{O}(T)$, as it needs to save batch parameters just as the traditional BN and additional $T$ values to scale the data at each time-step. But $T$ is usually small compared to $\mathcal{O}(CHW)$, this memory cost can be approximated by $\mathcal{O}(CHW)$, but we still keep the original one in order to show directly the difference of using the additional $T$ memory cost.
> - e) The total memory cost of our proposed TAB method is $\mathcal{O}(CHW) + \mathcal{O}(T)$, as it needs to scale the data as well as TEBN.
>
>
> **Computational Complexity:**
> - All the BN methods for SNNs need to deal with all the data spanning the time-dimension, no matter how to get the mean and variance,, all at once (e.g. tdBN and TEBN) or get that in a moving averaging way as in our proposed TAB method, the Computational Complexity is always $\mathcal{O}(TBCHW)$.
>
>
> ```
> [1] Youngeun Kim and Priyadarshini Panda. Revisiting batch normalization for training low-latency deep spiking neural networks from scratch. Frontiers in neuroscience, 2021.
> [2] Hanle Zheng, Yujie Wu, Lei Deng, Yifan Hu, and Guoqi Li. Going deeper with directly-trained larger spiking neural networks. In Proceedings of the AAAI conference on artificial intelligence, 2021.
> [3] Chaoteng Duan, Jianhao Ding, Shiyan Chen, Zhaofei Yu, and Tiejun Huang. Temporal effective batch normalization in spiking neural networks. In Advances in Neural Information Processing Systems, 2022.
> ```

---

> ### Author Response · Authors · 2023-11-22
> **Response to Reviewer vi7C (part 2/2)**
>
> > 3. Can you provide evidence that TAB solves TCS? I think the illustration of distribution is capable of demonstrating the solution.
>
>
> **Response 3:**
> Thank you for your insightful suggestion.
>
> - (1) In response, we have thoroughly investigated the distribution of presynaptic activities both before and after the application of TAB batch normalization.
>
> - (2) For our experimentation, we trained a 7-layer CNN model on DVS-CIFAR10 using the TAB method. The temporal distribution of normalization layers was visualized across various time-steps. Notably, all selected time-steps exhibited an almost identical distribution.
>
> - (3) This observation has shown evidence that TAB solves TCS, leads us to the conclusion that our TAB method effectively contributes to the adjustment of the current distribution in DVS data. These findings provide valuable insights into the adaptive capabilities of TAB in normalizing presynaptic activities in SNNs.

---

### Official Review · Reviewer_34HL · 2023-10-27

**Soundness:** 3 good
**Presentation:** 3 good
**Contribution:** 3 good
**Rating:** 8
**Confidence:** 5

**Summary:**

In the paper, the authors introduce a Temporal Accumulated Batch Normalization to address the temporal covariate shift issue by aligning with neuron dynamics and utilizing temporal accumulated statistics for data normalization.  They did experiments on CIFAR10, CIFAR100, and DVS-CIFAR10.

**Strengths:**

Ideas on low-bit precision neural networks are important and should be encouraged.
The paper focuses on temporal Covariate Shifts along the temporal dimension for SNN, which is an important problem in SNN.

**Weaknesses:**

The authors should provide an ablation study of the TAB, so we can know if the TAB can really increase the accuracy and how much the method improves.
Recent work all did experiments on ImageNet, while the paper ignore this.
Some references are missing.
[1] GLIF: A Unified Gated Leaky Integrate-and-Fire Neuron for Spiking Neural Networks.
[2] Reducing Information Loss for Spiking Neural Networks
[3] Surrogate Module Learning: Reduce the Gradient Error Accumulation in Training Spiking Neural Networks

**Questions:**

Please see weakness.

---

> ### Author Response · Authors · 2023-11-22
> **Response to Reviewer 34HL (part 1/2)**
>
> > 1. The authors should provide an ablation study of the TAB, so we can know if the TAB can really increase the accuracy and how much the method improves.
>
>
> **Response 1:**
> Thanks for the advice. In order to understand how the our TAB method increases the accuracy and how much the method improves, we have conducted ablation studies. We gradually added each part of TAB method and checked the performance change.
>
>
> - (1) **First, we conduct the ablation study on the effect of BN methods.**
>     As we are training SNNs, we need to deal with the time dimension, so we train the network without using any BN method. We conduct experiments using the same setting as TAB method, and compare the results with our TAB method. We can see that the models are not trained at all for all datasets with all network architectures.
>
>     | Data set  | Architecture | BN method  | Time-steps  |  Accuracy (%) |
>     |-----------|-----------|------------| ------------| ------------|
>     | CIFAR-10  | VGG-9 | No BN | 4 | 10 |
>     | CIFAR-10  | ResNet-19 | No BN  | 6 | 10 |
>     | CIFAR-10  | ResNet-19 | No BN  | 4 | 10 |
>     | CIFAR-10  | ResNet-19 | No BN  | 2 | 10 |
>     | CIFAR-100  | VGG-11 | No BN | 4 | 1 |
>     | CIFAR-100  | ResNet-19 | No BN | 6 | 1 |
>     | CIFAR-100  | ResNet-19 | No BN | 4 | 1 |
>     | CIFAR-100  | ResNet-19 | No BN | 2 | 1 |
>
>
>
> - (3) **Finally, we conduct the ablation study on the effect of learnable scaling weights $w[t]$** by removing the learnable scaling weights (i.e. setting $w[t]=1$) for all time-steps.
>     | Dataset  | Architecture | BN method  | Time-steps  |  Accuracy (%) |
>     |-----------|-----------|------------| ------------| ------------|
>     | CIFAR-10  | VGG-9 | No scaling, $w[t]=1$ | 4 | 91.93 |
>     | CIFAR-10  | ResNet-19 | No scaling, $w[t]=1$ | 6 | 94.78 |
>     | CIFAR-10  | ResNet-19 | No scaling, $w[t]=1$ | 4 | 94.3 |
>     | CIFAR-10  | ResNet-19 | No scaling, $w[t]=1$ | 2 | 93.83 |
>     | CIFAR-10-DVS  | CNN7 | No scaling, $w[t]=1$ | 4 | 74.2 |
>     | CIFAR-100  | VGG-11 | No scaling, $w[t]=1$ | 4 | 73.05 |
>     | CIFAR-100  | ResNet-19 | No scaling, $w[t]=1$ | 6 | 75.83 |
>     | CIFAR-100  | ResNet-19 | No scaling, $w[t]=1$ | 4 | 75.06 |
>     | CIFAR-100  | ResNet-19 | No scaling, $w[t]=1$ | 2 | 74.16 |
>
>     Results show that removing the scaling weights, the performance is worse than using the scaling weights.
>
>
> - (4) Finally, we show the results of our proposed TAB method for comparison.
>
>     | Dataset  | Architecture | BN method  | Time-steps  |  Accuracy (%) |
>     |-----------|-----------|------------| ------------| ------------|
>     | CIFAR-10  | VGG-9 | TAB | 4 | 93.41 |
>     | CIFAR-10  | ResNet-19 | TAB | 6 | 94.81 |
>     | CIFAR-10  | ResNet-19 | TAB | 4 | 94.76 |
>     | CIFAR-10  | ResNet-19 | TAB | 2 | 94.73 |
>     | CIFAR-100  | VGG-11 | TAB | 4 | 75.89 |
>     | CIFAR-100  | ResNet-19 | TAB | 6 | 76.82 |
>     | CIFAR-100  | ResNet-19 | TAB | 4 | 76.81 |
>     | CIFAR-100  | ResNet-19 | TAB | 2 | 76.31 |
>
>     Results show that the performance of our TAB method indeed increases when using batch normalization and including the scaling weights at the same time.

---

> ### Author Response · Authors · 2023-11-22
> **Response to Reviewer 34HL (part 2/2)**
>
> > 2. Recent work all did experiments on ImageNet, while the paper ignore this.
>
> **Response 2:**
> Thanks for your comments! We have conducted experiments on ResNet34 on ImageNet dataset.
> The following table summarises the results. Please refer to **To All Reviewers** for the details.
>
> - Due to the time limit of the rebuttal period, we have enabled to train 80 epochs of ResNet-34 on ImageNet with $T=2$, and 10 epochs for $T=4$.
> - Although we only train our model for 80 epochs for $T=2$, the TAB method achieves a $1.09\%$ increment on ResNet-34 with smaller latency over TEBN on $T=4$ (i.e. TAB=$ 65.38\% $ with $T=2$  v.s. TEBN=$ 64.29\%$ with $T=4$).
> - Even for only 10 epochs, the accuracy of our TAB method on ImageNet already reaches $ 65.98\%$ for $T=4$, with $1.7\%$ increment compared to TEBN with an accuracy of $64.29\%$ for $T=4$, which shows our TAB method is very promising.
>
>
> | Dataset  | Model  |  Methods |  Architecture  |  Time-steps | Accuracy (%) |
> | -------  | -------|------------| -------------| ------------|  ------------|
> | ImageNet | SPIKE-NORM [4] | ANN-to-SNN  |  ResNet-34 |  2500 | 69.96 |
> | ImageNet | RTS [5]  | ANN-to-SNN  |  VGG-16 |  16 | 55.80 |
> | ImageNet | QCFS [6] | ANN-to-SNN  | ResNet-34 |  16 | 59.35 |
> | ImageNet | SlipReLU [7] | ANN-to-SNN  | ResNet-34 |  32 | 66.61 |
> | ImageNet | SNNC-AP [8] | ANN-to-SNN  | ResNet-34 |  32 | 64.54 |
> | ImageNet | Hybrid Conversion [9] | Hybrid  | ResNet-34 |  250 | 61.48 |
> | ImageNet | TET [10] | Surrogate Gradient | Spiking-ResNet-34 |   6   | 64.79 |
> | ImageNet | tdBN [3] | Surrogate Gradient | ResNet-34 |   6   | 63.72 |
> | ImageNet | TEBN [11] | Surrogate Gradient | ResNet-34 |   4   | 64.29 |
> | ImageNet | TAB (ours) | Surrogate Gradient | ResNet-34  |   4   | 65.98 (10 epochs) |
> | ImageNet | TAB (ours) | Surrogate Gradient | ResNet-34  |   2   | 65.38 (80 epochs) |
>
>
> > 3. Some references are missing.
>
> **Response 3:**
> We will include all the listed references in the revised version of the main paper.

---

> > ### Comment · Reviewer_34HL · 2023-11-22
> >
> > Thanks for the response, my concern has been well addressed. I would like to increase my rating for the simple but effective method.

---

### Official Review · Reviewer_Y1dw · 2023-10-28

**Soundness:** 3 good
**Presentation:** 3 good
**Contribution:** 2 fair
**Rating:** 8
**Confidence:** 4

**Summary:**

The authors point out the challenging issue currently recognized in SNN-based models, which involves non-differentiable activation functions and temporally delayed accumulation of outputs over time. In this manuscript, the authors introduce TAB (Temporal Accumulated Batch Normalization), a novel SNN batch normalization method that addresses the temporal covariate shift issue by aligning with neuron dynamics (specifically the accumulated membrane potential) and utilizing temporal accumulated statistics for data normalization.

**Strengths:**

1. A very detailed theoretical analysis discusses the challenging problems of directly training SNN models through BN structures.
2. The authors introduce TAB, a novel SNN batch normalization method that addresses the temporal covariate shift issue by aligning with neuron dynamics and utilizing temporal accumulated statistics for data normalization.

**Weaknesses:**

1. The authors mention the phenomenon of covariance concept drift encountered by SNN models, so why don't they verify it on time-domain related datasets?
2. We can see that the improvement on the CIFAR-10 and CIFAR-100 datasets is very small, even just 0.6%. Can the current approach be regarded as a theoretically feasible strategy?
3. In addition, the authors have not looked at the impact of other network skeletons, such as ResNet34.
4. Another issue reviewer is concerned about is the relationship between TAB and neuron dynamics. Does the exponential nonlinear operation of the SNN-based model on the time coefficient lead to an error in approximating a first-order linear ODE?

Feedback: The author added a large number of revisions in the limited time, I think I have no new concerns.

**Questions:**

Please see details of weaknesses.

---

> ### Author Response · Authors · 2023-11-21
> **Response to Reviewer Y1dw (part 1/2)**
>
> > 1. The authors mention the phenomenon of covariance concept drift encountered by SNN models, so why don't they verify it on time-domain related datasets?
>
>
> **Response 1:**
> Thanks for your advice.
>
> - (a) The phenomenon of covariance concept drift encountered by SNN models is raised directly by the LIF spiking neuron model and its neuron dynamics, where each neuron maintains its membrane potential dynamics $u(t)$ over time, "integrates" the received input with a leakage, and fires a spike if the accumulated membrane potential value exceeds a threshold. SNNs use binary spike trains to transmit information between layers. The spike train is a series of 0's and 1's like `spike-train=[0,0,0,1,0,0,1]` with 0 indicating the neuron is silent, and 1 indicating the neuron is active.
>
> - (2) The neuron dynamics refer to the changes in the membrane potential of a neuron over time as it integrates input signals and generates spikes. It can be formulated as a first-order differential equation (ODE),
> $\tau \frac{d u(t)}{d t} = -u(t) + I(t) $.
>
>
> - (3) Within SNNs, synaptic currents $I(t)$ are sequentially fed into spiking neurons, with spike-triggered asynchronous currents accumulating in the membrane potential. Whenever this accumulated membrane potential exceeds a threshold, a spike is generated. This temporal dependency on membrane accumulation has the potential to amplify the internal covariate shift across the temporal domain which is termed the Temporal Covariate Shift (TCS) phenomenon.
>
>
> - (4) In ANNs, the internal covariate shift (ICS) is incurred due to changes in the distribution of layer inputs caused by updates of preceding layers. In SNNs, this covariate shift does not only come from the layer updating, but also the temporal updating. The Temporal Covariate Shift phenomenon is due to updates of preceding layers and prior time-steps, which specially transpire along the additional temporal dimension compared to ANNs. Therefore, the intertwining of this temporal dependency of membrane potential dynamics $u(t)$ with the TCS phenomenon, presents a significant challenge in direct training of SNNs.
>
>
> > 2. We can see that the improvement on the CIFAR-10 and CIFAR-100 datasets is very small, even just 0.6%. Can the current approach be regarded as a theoretically feasible strategy?
>
>
> **Response 2:**
>
> - (1) Yes, our proposed TAB approach provides a theoretically feasible strategy of Batch Normalization in training SNNs, as it addresses the temporal covariate shift issue by aligning with neuron dynamics and utilizing temporal accumulated statistics for data normalization. A detailed theoretical analysis has been provided in the main paper.
>
> - (2) Meanwhile, as for the improvement of our proposed TAB method with other SOTA methods, for example on the CIFAR-10 and CIFAR-100 datasets, results of the statistical hypothesis have shown that there is a statistically significant improvement of our TAB method over other SOTA methods.
>
>
> - (3) As suggested by *Reviewer ZLsc*, we have conducted a statistical hypothesis testing to check if the improvement of our TAB method is statistically significant over other SOTA methods. We have also done the Wilcoxon signed-rank test. Results show p-value=0.000244140625 which is less than 0.001 (the critical value), indicating that there is a significant difference in the performance between TAB and TEBN, statistically. Please check ***Response to All Reviewers (part 1/3)*** for the details.
>
>
> - (4) We also checked the rank of the performance of our TAB method and TEBN, our TAB method always ranks 1 among all the paired TAB-TEBN performance results.

---

> ### Author Response · Authors · 2023-11-21
> **Response to Reviewer Y1dw (part 2/2)**
>
> > 3. In addition, the authors have not looked at the impact of other network skeletons, such as ResNet34.
>
> **Response 3:**
> - (1) Thanks for your comments! We have provided the experiment results of ResNet34 on the ImageNet dataset. The following table summarizes the results.
> Please refer to ***Response to All Reviewers (part 3/3)*** for the details.
>
> - (2) Due to the time limit of the rebuttal period, we have enabled training 80 epochs of ResNet-34 on ImageNet with $T=2$, and 10 epochs for $T=4$.
> Although we only train our model for 80 epochs for $T=2$, the TAB method achieves a $1.09\%$ increment on ResNet-34 with smaller latency over TEBN on $T=4$ (i.e. TAB=$ 65.38\% $ with $T=2$  v.s. TEBN=$ 64.29\%$ with $T=4$). Even for only 10 epochs, the accuracy of our TAB method on ImageNet already reaches $ 65.98\%$ for $T=4$, with $1.7\%$ increment compared to TEBN with an accuracy of $64.29\%$ for $T=4$, which shows our TAB method is very promising.
>
>
> | Dataset  | Model  |  Methods |  Architecture  |  Time-steps | Accuracy (%) |
> | -------  | -------|------------| -------------| ------------|  ------------|
> | ImageNet | SPIKE-NORM [4] | ANN-to-SNN  |  ResNet-34 |  2500 | 69.96 |
> | ImageNet | RTS [5]  | ANN-to-SNN  |  VGG-16 |  16 | 55.80 |
> | ImageNet | QCFS [6] | ANN-to-SNN  | ResNet-34 |  16 | 59.35 |
> | ImageNet | SlipReLU [7] | ANN-to-SNN  | ResNet-34 |  32 | 66.61 |
> | ImageNet | SNNC-AP [8] | ANN-to-SNN  | ResNet-34 |  32 | 64.54 |
> | ImageNet | Hybrid Conversion [9] | Hybrid  | ResNet-34 |  250 | 61.48 |
> | ImageNet | TET [10] | Surrogate Gradient | Spiking-ResNet-34 |   6   | 64.79 |
> | ImageNet | tdBN [3] | Surrogate Gradient | ResNet-34 |   6   | 63.72 |
> | ImageNet | TEBN [11] | Surrogate Gradient | ResNet-34 |   4   | 64.29 |
> | ImageNet | TAB (ours) | Surrogate Gradient | ResNet-34  |   4   | 65.98 (10 epochs) |
> | ImageNet | TAB (ours) | Surrogate Gradient | ResNet-34  |   2   | 65.38 (80 epochs) |
> |
>
>
>
>
>
> > 4. Another issue reviewer is concerned about is the relationship between TAB and neuron dynamics. Does the exponential nonlinear operation of the SNN-based model on the time coefficient lead to an error in approximating a first-order linear ODE?
>
>
> **Response 4:**
> - (1) Yes, but there will be an approximation error, but this is due to the discretization process.
>
> - (2) The analytical closed-form solution for the first-order Initial Value Problem of the LIF dynamics ODE is
>     $$
>         U(t) =  \exp{(-\frac{t}{\tau}) } \left( \int_0^t \frac{R}{\tau} I(s) \exp{ (\frac{s}{\tau}) } ds + U_0 \right) \ .
>     $$
>     When the neuron initiates at the value $U_0$ with no further input, i.e., $I(t)=0$, the closed-form solution of the ODE shows that the membrane potential $U(t)$ will start at $U_0$ and exponentially decay with a time constant $\tau$, $U(t) = U_0 \exp{ (-\frac{t}{\tau}) }$.
>     However, for the exponential term, it is absorbed in $\lambda$, as $\lambda = \frac{U(t+\Delta t)}{ U(t)} = \frac{U_0 \exp{ (-\frac{t+\Delta t}{\tau}) }}{ U_0 \exp{ (-\frac{t}{\tau}) }} = \exp{ (-\frac{\Delta t}{\tau}) }$. This relationship enables us to formulate the discretization scheme $U[t+1] = \lambda U[t]$.
>
>
> - (3) Through further rigorous derivation, we derive another equivalent form of the closed-form solution for the LIF dynamics,
>     $$
>     U(t) = (U_0 -R I_0) \exp{ (-\frac{t}{\tau}) } + R I(t) -\int_0^t R \exp{(\frac{s-t}{\tau}) } dI(s)
>     $$
>     The commonly used discrete LIF model, as denoted by $U[t] = \lambda U[t-1] + X[t]$, is derived from the first two terms of the discretization version of the closed-form solution.
>
> - (4) Recalling our TAB method normalizes data utilizing temporal accumulated batch statistics ($\mu_{1:t}, \sigma^2_{1:t}$) across an expanding window $[1, t]$, where $\mu_{1:t}$ and $\sigma^2_{1:t}$ represent the temporal accumulated information up to time-step $[t]$. Computation of these temporal accumulated statistics is performed dynamically, employing a moving averaging approach, obviating the need to store batch data from all previous time-steps.
> The utilization of the temporal accumulated batch statistics aligns well with the accumulation mechanism of the membrane potential through the above equation.

---

> > ### Comment · Reviewer_Y1dw · 2023-11-22
> > **Successfully addressed all my concerns!**
> >
> > The author added a large number of revisions in the limited time, I think I have no new concerns, and I am more than willing to improve my score.

---

### Official Review · Reviewer_ZLsc · 2023-10-30

**Soundness:** 3 good
**Presentation:** 3 good
**Contribution:** 3 good
**Rating:** 6
**Confidence:** 3

**Summary:**

In Artificial Neural Networks (ANNs), Internal co-variate shift denotes alterations in the input distribution due to preceding layer updates. In Spiking Neural Networks (SNNs), the Temporal Covariate Shift (TCS) problem arises from both previous layer updates and prior time steps, extending along the temporal dimension. Within SNNs,  neurons accumulate incoming spikes in their membrane potential over time and only generate a spike when this accumulation surpasses a threshold, staying inactive otherwise within the current time-step.
Hence, the accumulation of synaptic currents in the membrane potential is affected by temporal dependencies, which could magnify the internal covariate shift as time progresses. The central concept of this study revolves around addressing the challenge of applying batch normalization during SNN training while considering the temporal data dependencies and the issue of temporal covariate shift. TAB (Temporal Accumulated Batch Normalization) is proposed which closely follows neuron dynamics, especially the accumulated membrane potential, enhancing the accuracy of batch statistics. This alignment establishes a natural link between the behavior of neurons and the application of batch normalization in Spiking Neural Networks (SNNs). TAB uses Temporal Accumulated Statistics (dynamically using a moving averaging approach) for data normalization, introducing learnable weights to differentiate the impact of each time-step on the final result. The study also provides a theoretical connection between TAB method and the neural dynamics.

**Strengths:**

The paper is well written and easy to read. The authors presented an extensive set of experimental results to evaluate the proposed approach. The theoretical connection between the underlying neural dynamics and the propose normalization technique is very well explained and presented.

**Weaknesses:**

The experimental results does not show a significant improvement over SOTA for example TEBN, I would suggest including a demsar plot to show the improvement also explain fairly why the improvement is very small. Taking into account all the added complexity regarding trainable weights for sequential neurons, etc.

**Questions:**

You have used trainable weights for the each point in the neurons sequence, however it is not explain well how the weights for a new data can be used. Imagine you see a shift in the data distribution are these weights still valid?

---

> ### Author Response · Authors · 2023-11-21
> **Response to Reviewer ZLsc (part 1/2)**
>
> > Q1. regarding performance improvement over other SOTA methods (for example TEBN), suggest including a demsar plot to show the improvement
>
>
> **Response 1:**
> Thank you for your insightful comment! we have included a demsar plot regarding our TAB method over other SOTA methods, refer [here](https://anonymous.4open.science/r/ICLR2024_TAB_Rebuttal-BF2C/cd-diagram.pdf) for the plot.
>
> - (1) **For the demsar plot**, the difference in the average ranking on the axis of two algorithms is greater than the CD (critical difference) value, indicating that there is a statistically significant difference between our TAB method and the TEBN method. And our TAB method's performance is always ranking the first, so our TAB has a consistent improvement among all the results.
>
>
> - (2) **For the statistical hypothesis test**, we have done the Wilcoxon signed-rank test to see whether the performance of our proposed TAB method is statistically different from other SOTA methods. Results show p-value=0.000244140625 which is less than 0.001 (the critical value), indicating that there is a significant difference in the performance between TAB and TEBN.
>
> - (3) As among other SOTA methods, TEBN performs the best, so TEBN is seen as the most competitive method to TAB. We have conducted a statistical hypothesis test, the Wilcoxon signed-rank test on the paired TAB-TEBN performance results. By employing Wilcoxon signed-rank test, we have obtained p-value=0.000244140625 which is less than 0.001 (the critical value), indicating that there is a significant difference in the performance between TAB and TEBN.
>
> - (4) We also checked the rank of the performance of our TAB method and TEBN, our TAB method always ranks 1 among all the paired TAB-TEBN performance results.
>
>
> > Details:
>
> - (1). For the statistical hypothesis test
>
>     The Wilcoxon signed-rank test is a non-parametric statistical test used to determine whether there is a difference between paired observations. As the Wilcoxon signed-rank test does not assume normality in the data and can be widely used, here we use Wilcoxon signed-rank test to understand whether there is a difference in the performance from different BN methods, typically between our proposed TAB method and its most competitor TEBN. Here are the Null Hypothesis and the Alternative Hypothesis.
>     ```
>     Null Hypothesis (H0): there is no difference in the performance between TAB and TEBN.
>
>     Alternative Hypothesis (H1): There is a significant difference in the performance between TAB and TEBN.
>
>     ```
>
>     We collect the observed paired data, for example, for VGG-9 on cifar10 with time-step=4, the results of [TAB, TEBN] are [92.81, 93.41]. From table 2 and table 3 in the main paper, we have 13 such paired data.
>
>
> - (2). For the demsar plot
>
>     The value on the x-axis is the CD (critical difference) value. The value on the x-axis is the average ranking of each algorithm on multiple data sets.
>     If the difference in the average ranking of the two algorithms is greater than the CD (critical difference) value, it indicates that there is a statistically significant difference between these two algorithms.

---

> > ### Comment · Reviewer_ZLsc · 2023-11-22
> >
> > Thanks for clarifications. However, for demsar plot, it makes sense to put all the comparison approaches not only TEBN.

---

> > > ### Author Response · Authors · 2023-11-22
> > >
> > > - Thank you for your suggestion. Including all comparison approaches totally makes sense. But due to limited paired data (only 3 comparisons for other methods), statistical testing might not be possible. We have presented the available data comprehensively, where we have 13 paired data with TEBN and TEBN is the most competitive method compared to our TAB method. That is why we have shown the demsar plot with TEBN.
> > >
> > > - Results TAB consistently outperformed TEBN, showing a statistically significant difference between our TAB method and the TEBN method. Our TAB consistently ranked first, indicating a reliable improvement. Thanks for your advice, feel free to discuss any further questions.

---

> ### Author Response · Authors · 2023-11-21
> **Response to Reviewer ZLsc (part 2/2)**
>
> > Q2. Taking into account all the added complexity regarding trainable weights for sequential neurons, etc.
>
> **Response 2:**
> Thanks for your comments! We have provided a thorough analysis and a detailed explanation for this concern.
> The following table summarizes the Complexity Cost of different BN methods, including computational complexity and Memory Cost.
> Please refer to ***Response to All Reviewers (part 2/3)*** for the details.
>
>
> | BN methods | Computational Complexity | Memory Cost |
> | -------|------------| -------------|
> | Conventional BN | $\mathcal{O}(BCHW) $  |  $\mathcal{O}(CHW)$ |
> | BNTT (Kim & Panda, 2021) | $\mathcal{O}(TBCHW)$ |  $\mathcal{O}(TCHW)$ |
> | tdBN (Zheng et al., 2021) | $\mathcal{O}(TBCHW)$ | $\mathcal{O}(T)$ |
> | TEBN (Duan et al., 2022) | $\mathcal{O}(TBCHW)$  |  $\mathcal{O}(CHW) + \mathcal{O}(T)$ |
> | TAB (ours) | $\mathcal{O}(TBCHW)$ |  $\mathcal{O}(CHW) + \mathcal{O}(T)$ |
> |
>
>
> >  Q3. You have used trainable weights for the each point in the neurons sequence. Explain how the trainable weights for a new data can be used. Imagine you see a shift in the data distribution are these weights still valid?
>
> **Response 3:**
> - (1) The trainable weights are introduced in the TAB normalization and in the implementation, they will be used in the TAB-layer in the same way we use them in the conventional BN layers. In conventional BN layers, the batch-normalized inputs are scaled and shifted using learnable/trainable parameters $\gamma, \beta$ to preserve model expressiveness,
> $BN(x_i) = \gamma \hat{x}_i + \beta $, and $\hat{x}_i = \frac{x_i - \mu}{\sqrt{\sigma^2 + \epsilon}} $.
>
> - (2) For the network training phase, the trainable BN weights $\gamma, \beta$ need to be learned together with the network's weights $W$. When there is a shift in the data distribution, the network adjusts the trainable weights $\gamma, \beta$ so as to alleviate the shifting phenomena in the data distribution.
>
> - (3) Our TAB method employs the same normalization logic to the trainable weights $\gamma, \beta$, with an additional trainable scaling parameter $w[t]$ to scale along the temporal dimension, so the final parameters are $\hat\gamma = w[t]\gamma, \hat\beta=w[t]\beta$. They need to be learned in the TAB-layer.

---

### Author Response · Authors · 2023-11-21
**# Response to All Reviewers (part 1/3)**

**We sincerely thank all reviewers for their insightful and constructive feedback!**
We have carefully answered the relevant questions raised by every reviewer. We sincerely hope that you can make brand-new evaluations for our work.

Here we would like to address the general concerns surrounding the issues of "significance of the performance improvement", "the complexity analysis", as well as "Experiments on ResNet34", as raised by most reviewers.


> 1. Is there a significant improvement of the proposed TAB over other SOTA (for example TEBN)?

- (a) Yes. Results of the statistical hypothesis test show a statistically significant improvement of the proposed TAB over other SOTA.

- (b) **For the statistical hypothesis test**, we have done the Wilcoxon signed-rank test to see whether the performance of our proposed TAB method is statistically different from other SOTA methods. Results show p-value=0.000244140625 which is less than 0.001 (the critical value), indicating that there is a significant difference in the performance between TAB and TEBN.

- (c) **For the demsar plot**, the difference in the average ranking on the axis of two algorithms is greater than the CD (critical difference) value, indicating that there is a statistically significant difference between our TAB method and the TEBN method. And our TAB method's performance is always ranking the first, so our TAB has a consistent improvement among all the results. Check the demsar plot here https://anonymous.4open.science/r/ICLR2024_TAB_Rebuttal-BF2C/cd-diagram.pdf.


- (d) We also checked the rank of the performance of our TAB method and TEBN, our TAB method always ranks 1 among all the paired TAB-TEBN performance results.

---

### Author Response · Authors · 2023-11-21
**# Response to All Reviewers (part 2/3)**

> 2. Complexity analysis

The Table summarizes the Complexity Cost of different BN methods, including computational complexity and Memory Cost.

| BN methods | Computational Complexity | Memory Cost |
| -------|------------| -------------|
| Conventional BN | $\mathcal{O}(BCHW) $  |  $\mathcal{O}(CHW)$ |
| BNTT [1] | $\mathcal{O}(TBCHW)$ |  $\mathcal{O}(TCHW)$ |
| tdBN [2] | $\mathcal{O}(TBCHW)$ | $\mathcal{O}(CHW)$ |
| TEBN [3] | $\mathcal{O}(TBCHW)$  |  $\mathcal{O}(CHW) + \mathcal{O}(T)$ |
| TAB (ours) | $\mathcal{O}(TBCHW)$ |  $\mathcal{O}(CHW) + \mathcal{O}(T)$ |


- (1) We have obtained this Complexity analysis based on the statistics and parameters of different BN methods in Table 1 in the main paper. We use here T to refer to the number of time-steps, B to refer to the batch size, C,H,W are the number of channels, height, and width. So the input at time-step t, $X_t$, to the normalization layer in SNNs has a dimension of $\mathbf{R}^{C \times H \times W}$.

- (2) In the conventional BN method in ANNs, there is no time dimension, therefore, the traditional BN method in ANNs has the computational complexity of $\mathcal{O}(BCHW) $, and a Memory Cost of $\mathcal{O}(CHW)$ to save the mean and variance which have the same dimension as $X$.


**Memory Cost:**
- a) As BNTT independently normalizes data at each time-step, it needs $\mathcal{O}(TCHW)$ space to save the mean and variance at each time-step, which is T times more than the traditional BN method, tdBN, and TEBN.
- b) The tdBN and TEBN jointly normalize data across all time-steps, so they only need $\mathcal{O}(CHW)$ space to save one overall mean and variance which is the same as the traditional BN method for saving mean and variance.
- c) The total memory cost of tdBN is $\mathcal{O}(CHW)$, as it needs only shared overall batch parameters just as the traditional BN, so there is no additional memory cost for saving the batch parameters.
- d) The total memory cost of TEBN is $\mathcal{O}(CHW) + \mathcal{O}(T)$, as it needs to save batch parameters just as the traditional BN and additional $T$ values to scale the data at each time-step. But $T$ is usually small compared to $\mathcal{O}(CHW)$, this memory cost can be approximated by $\mathcal{O}(CHW)$, but we still keep the original one in order to show directly the difference of using the additional $T$ memory cost.
- e) The total memory cost of our proposed TAB method is $\mathcal{O}(CHW) + \mathcal{O}(T)$, as it needs to scale the data as well as TEBN.


**Computational Complexity:**
- All the BN methods for SNNs need to deal with all the data spanning the time-dimension, no matter how to get the mean and variance, all at once (e.g. tdBN and TEBN) or get that in a moving averaging way as in our proposed TAB method, the Computational Complexity is always $\mathcal{O}(TBCHW)$.


```
[1] Youngeun Kim and Priyadarshini Panda. Revisiting batch normalization for training low-latency deep spiking neural networks from scratch. Frontiers in neuroscience, 2021.
[2] Hanle Zheng, Yujie Wu, Lei Deng, Yifan Hu, and Guoqi Li. Going deeper with directly-trained larger spiking neural networks. In Proceedings of the AAAI conference on artificial intelligence, 2021.
[3] Chaoteng Duan, Jianhao Ding, Shiyan Chen, Zhaofei Yu, and Tiejun Huang. Temporal effective batch normalization in spiking neural networks. In Advances in Neural Information Processing Systems, 2022.
```

---

### Author Response · Authors · 2023-11-21
**Response to All Reviewers (part 3/3)**

### 3. Experiments on other network skeletons, such as ResNet34.

We have conducted experiments of ResNet34 on the ImageNet dataset. Here are the results.


| Dataset  | Model  |  Methods |  Architecture  |  Time-steps | Accuracy (%) |
| -------  | -------|------------| -------------| ------------|  ------------|
| ImageNet | SPIKE-NORM [4] | ANN-to-SNN  |  ResNet-34 |  2500 | 69.96 |
| ImageNet | RTS [5]  | ANN-to-SNN  |  VGG-16 |  16 | 55.80 |
| ImageNet | QCFS [6] | ANN-to-SNN  | ResNet-34 |  16 | 59.35 |
| ImageNet | SlipReLU [7] | ANN-to-SNN  | ResNet-34 |  32 | 66.61 |
| ImageNet | SNNC-AP [8] | ANN-to-SNN  | ResNet-34 |  32 | 64.54 |
| ImageNet | Hybrid Conversion [9] | Hybrid  | ResNet-34 |  250 | 61.48 |
| ImageNet | TET [10] | Surrogate Gradient | Spiking-ResNet-34 |   6   | 64.79 |
| ImageNet | tdBN [3] | Surrogate Gradient | ResNet-34 |   6   | 63.72 |
| ImageNet | TEBN [11] | Surrogate Gradient | ResNet-34 |   4   | 64.29 |
| ImageNet | TAB (ours) | Surrogate Gradient | ResNet-34  |   4   | 65.98 (10 epochs) |
| ImageNet | TAB (ours) | Surrogate Gradient | ResNet-34  |   2   | 65.38 (80 epochs) |
|


- (a) We verify our TAB method on the ImageNet dataset. ImageNet dataset [1] has more than 1250k training images and 50k test images. The training set of ImageNet provides 1.28k training samples for each label. We use the standard pre-processing and augmentation for training data [2]. The test data is directly centered and cropped to $224 \times 224$.

- (b) We chose the representative architecture ResNet-34 to verify our TAB method on ImageNet. The network is trained with AdamW optimizer. The initial learning rate is set to 0.00002 and weight decay is set to 0.02.
We train the model on an NVIDIA RTX A6000 machine with 4 GPUs using a batch size of 24 per GPU. Following previous studies [3], to avoid biased statistics, we synchronize the batch mean and variance across each device. Results are demonstrated in the following Table.


- (c) Due to the time limit of the rebuttal period, we have enabled to training 80 epochs of ResNet-34 on ImageNet with $T=2$, and 10 epochs for $T=4$.
Although we only train our model for 80 epochs for $T=2$, the TAB method achieves a $1.09\%$ increment on ResNet-34 with smaller latency over TEBN on $T=4$ (i.e. TAB=$ 65.38\% $ with $T=2$  v.s. TEBN=$ 64.29\%$ with $T=4$). Even for only 10 epochs, the accuracy of our TAB method on ImageNet already reaches $ 65.98\%$ for $T=4$, with $1.7\%$ increment compared to TEBN with an accuracy of $64.29\%$ for $T=4$, which shows our TAB method is very promising.


```
[1] Jia Deng, Wei Dong, Richard Socher, Li-Jia Li, Kai Li, and Li Fei-Fei. Imagenet: A large-scale hierarchical image database. In 2009 IEEE conference on computer vision and pattern recognition, 2009.
[2] Kaiming He, Xiangyu Zhang, Shaoqing Ren, and Jian Sun. Deep residual learning for image recognition. In Proceedings of the IEEE conference on computer vision and pattern recognition, 2016.
[3] Hanle Zheng, Yujie Wu, Lei Deng, Yifan Hu, and Guoqi Li. Going deeper with directly-trained larger spiking neural networks. In Proceedings of the AAAI conference on artificial intelligence, 2021.
[4] Abhronil Sengupta, Yuting Ye, Robert Wang, Chiao Liu, and Kaushik Roy. Going deeper in spiking neural networks: VGG and residual architectures. Frontiers in neuroscience, 2019.
[5] Shikuang Deng and Shi Gu. Optimal conversion of conventional artificial neural networks to spiking neural networks. In International Conference on Learning Representations, 2021.
[6] Tong Bu, Wei Fang, Jianhao Ding, PengLin Dai, Zhaofei Yu, and Tiejun Huang. Optimal ANN-SNN conversion for high-accuracy and ultra-low-latency spiking neural networks. In International Conference on Learning Representations, 2022.
[7] Haiyan Jiang, Srinivas Anumasa, Giulia De Masi, Huan Xiong, and Bin Gu. A unified optimization framework of ann-snn conversion: Towards optimal mapping from activation values to firing rates. International Conference on Machine Learning, 2023.
[8] Yuhang Li, Shikuang Deng, Xin Dong, Ruihao Gong, and Shi Gu. A free lunch from ANN: Towards efficient, accurate spiking neural networks calibration. In International Conference on Machine Learning, 2021.
[9] Nitin Rathi, Gopalakrishnan Srinivasan, Priyadarshini Panda, and Kaushik Roy. Enabling deep spiking neural networks with hybrid conversion and spike timing dependent backpropagation. In International Conference on Learning Representations, 2020.
[10] Shikuang Deng, Yuhang Li, Shanghang Zhang, and Shi Gu. Temporal efficient training of spiking neural network via gradient re-weighting. In International Conference on Learning Representations, 2022.
[11] Chaoteng Duan, Jianhao Ding, Shiyan Chen, Zhaofei Yu, and Tiejun Huang. Temporal effective batch normalization in spiking neural networks. In Advances in Neural Information Processing Systems, 2022.
```

---

### Meta-Review · Area_Chair_141T · 2023-12-11

**Metareview:**

The paper introduces the TAB method, addressing the Temporal Covariate Shift in SNNs, a notable contribution to the field. While the reviewers commend the comprehensive experiments and theoretical backing, they express concerns over modest performance improvements and complexity. The authors provide persuasive rebuttals, including additional ImageNet results. Despite some limitations, the paper’s strengths in innovation and thorough experimental validation make it a valuable addition to the conference.

**Justification For Why Not Higher Score:**

The overall impact and broad appeal of the paper does not meet the typical criteria for spotlight papers at ICLR.

**Justification For Why Not Lower Score:**

The paper is above the acceptance threshold.

---

### Decision · Program_Chairs · 2024-01-16

Accept (poster)